# Integration of multiple electronic components on a microfibre towards an emerging electronic textile platform

Sunbin Hwang[1,8], Minji Kang[2,7,8], Aram Lee[1], Sukang Bae [1], Seoung-Ki Lee[3], Sang Hyun Lee [4], Takhee Lee [5], Gunuk Wang [6] & Tae-Wook Kim [2✉]

Electronic fibres have been considered one of the desired device platforms due to their dimensional compatibility with fabrics by weaving with yarns. However, a precise connecting process between each electronic fibre is essential to configure the desired electronic circuits or systems. Here, we present an integrated electronic fibre platform by fabricating electronic devices onto a one-dimensional microfibre substrate. Electronic components such as transistors, inverters, ring oscillators, and thermocouples are integrated together onto the outer surface of a fibre substrate with precise semiconductor and electrode patterns. Our results show that electronic components can be integrated on a single fibre with reliable operation. We evaluate the electronic properties of the chip on the fibre as a multifunctional electronic textile platform by testing their switching and data processing, as well as sensing or transducing units for detecting optical/thermal signals. The demonstration of the electronic fibre suggests significant proof of concepts for the realization of high performance with wearable electronic textile systems.

[1] Functional Composite Materials Research Center, Korea Institute of Science and Technology, Wanju-gun, Jeollabuk-do 55324, Republic of Korea. [2] Department of Flexible and Printable Electronics, LANL-JBNU Engineering Institute-Korea, Jeonbuk National University, 567 Baekje-daero, Deokjin-gu, Jeonju 54896, Republic of Korea. [3] School of Materials Science and Engineering, Pusan National University, 2, Busandaehak-ro-63-beon-gil, Geumjeong-gu, Busan 46241, Republic of Korea. [4] School of Chemical Engineering, Chonnam National University, 77 Yongbong-ro, Buk-gu, Gwangju 61186, Republic of Korea. [5] Department of Physics and Astronomy, Institute of Applied Physics, Seoul National University, Seoul 08826, Republic of Korea. [6] KU-KIST Graduate School of Converging Science and Technology, Korea University, 145 Anam-ro, Seongbuk-gu, Seoul 02841, Republic of Korea. [7] Present address: Chemical Materials Solutions Center, Korea Research Institute of Chemical Technology, 141 Gajeong-ro, Yuseong-gu, Daejeon 34114, Republic of Korea. [8] These authors contributed equally: Sunbin Hwang, Minji Kang. ✉email: twk@jbnu.ac.kr

Fibre electronics are of considerable interest for wearable applications and smart textiles, and they can facilitate communication and the interaction between humans and surroundings[1–3]. As a basic element of functional textiles, the one-dimensional (1D) form of thread-like fibres offers high flexibility, isotropic deformations, breathability, and lightweight in fabric structures[4,5]. The 1D functional fibres can be further processed into two-dimensional (2D) textile and three-dimensional (3D) yarn configurations through traditional textile engineering techniques, such as twisting, weaving, sewing, knitting, knotting, and interlacing[5,6]. Owing to such intrinsic merits, in recent years, fibre-based device components that perform optoelectronic functions, such as health/environmental monitoring, displays, sensing, energy harvesting, energy storage, electromagnetic shielding, and information processing, have been integrated directly into fabrics to demonstrate futuristic clothes[7–14].

The existing electronic fibre platforms are generally composed of only one type of electronic component with a single function on a fibre substrate that is attributed to all around wrapping of an active layer on the entire fibre without patterning at the desired area on the surface of the fibre during the manufacturing process. Moreover, a precise connecting process between each electronic fibre is essential to configure the desired electronic circuits or systems into the 2D textile while minimizing the degradation of the device performance[15]. Although assembly of those functional fibres can be used for recording, detecting, and readout data sequentially, similar to conventional integrated circuits and multifunctional devices on 2D wafers, both limitations on scaling down and difficulty in the configuration of the electronic circuit remain major obstacles for the implementation of practical electronic fibre systems. First, many complex and functional connections are generated from large-scale integration (LSI), and thus, reducing wiring, such as conductive threads, is considered a bottleneck for further development. Second, the areal density of the device should be increased by introducing a specifically designed architecture or process[16]. From this point of view, it is highly necessary to develop compact and miniaturized electronic systems that are capable of working on a single fibre. To impart multiple functions to the textile, the methods of inserting small electronic components into a fibre strand or yarn have been considered emerging candidates, enabling the implementation of a thermally drawn digital fibre and e-yarn[17–19]. However, a limitation to the thermal drawing approach and the mounting of small components on the top surface of a filament is the low device density. A strategy to fabricate a high-density electronic microfibre possessing multiple electronic components and circuits as well as maintaining excellent electrical performance has not yet been reported.

In this work, we present an electronic fibre platform that enables LSI of electronic device components on the surface of a 1D fibre, defined as a monofilament with a diameter of 150 μm (Fig. 1a). By using high-resolution maskless photolithography with a capillary tube-assisted coating method[20], multiple miniaturized device units are integrated onto a very narrow and thin fibre surface. As a proof-of-concept demonstration, basic electronic devices (field-effect transistors, inverters, and ring oscillators) and sensors (photodetectors, signal transducer, and distributed temperature sensors consisting of thermocouples) are fabricated onto the two different sides of the rectangular fibre. The chip on a fibre exhibits various electronic functions (UV detection and switching electrical signals in a single transistor, symmetric input/output behaviour in the n-type inverter, oscillation characteristics of 5-stage ring oscillator) and thermal sensing performance. We believe that our approach is one of the big steps to implement a high-density electronic fibre platform for integrated electronic textiles.

## Results

**Assembly of multiple electronic systems on a microfibre.** The assembly of multiple electronic systems on a microfibre, illustrated in Fig. 1A, B, consists of two different electronic parts: basic optoelectronic elements and a temperature sensor. The electronic components are integrated on the surface of a ten-centimetre-long cuboid shape monofilament with a diameter of 150 μm. As a proof of concept to implement direct assembly of electronic systems on a microfibre, our electronic fibre has relatively low integration density compared to conventional electronic systems on 2D wafers. However, by further scaling down each electronic part on a microfibre, it may be possible to implement high-density electronic fibres similar to those of conventional semiconductor devices. This implies that our electronic fibre platform can be considered one of the potential emerging electronic fibres. A square-shaped microfibre made of fused silica was employed as a transparent and flexible substrate. Although the silica-based fibre has relatively lower flexibility than polymeric substrates, it can sustain high process temperature of 1100 °C without melting, allowing high-performance inorganic electrical materials to be deposited on it. The microfibre substrate also features a 3D geometric shape including four planar faces throughout the length of the fibre that enables a higher integration density. The most efficient way to increase the density of electronic components is to use the entire circumference of fibre. In order to investigate the feasibility of this, we integrated electronic components on two different sides of the square sectional fibre. A transistor, an inverter, and a ring oscillator (RO) based on an indium gallium zinc oxide (IGZO) metal oxide semiconductor (MOS) are placed on the top surface of the fibre while the temperature sensor is built onto the side of the fibre. To demonstrate the whole device, we exploited both a capillary tube-assisted coating (CTAC) method and high-resolution maskless photolithography, which is able to quickly fabricate patterned metal electrodes onto the two different sides of a thin and narrow monofilament substrate[20,21]. The CTAC process has the potential to be compatible with a reel-to-reel coating process, which is an efficient way to minimize material waste and allows fine control of photoresist (PR) film thickness by adjusting coating speed and solution concentration[20]. Cross-sectional scanning electron microscopy (SEM) images indicate that the CTAC-processed PR film uniformly covered the entire outer surface of the fibre, and the thickness of the PR layer was estimated to be approximately 2 μm (Supplementary Fig. 1). After coating and baking the PR film on the fibre, a laser pattern generator was employed to quickly expose the PR along with the electrode pattern (Supplementary Fig. 1). Maskless lithography directly transfers the design patterns onto the fibre substrate without utilizing a photomask and enables exact positioning of the electrode patterns to fabricate desired electronic devices at arbitrary locations on nonplanar substrates, as shown in Supplementary Fig. 2[21]. Experimental details (the deposition of metal thin films and wet etching through photolithography) and the electrode patterns formed on the fibre are also described in Supplementary Fig. 3. Figures 1C, D show a SEM image and a photograph of the entire device fabricated on the microfibre substrate. An integrated electronic fibre (length: 10 cm) containing approximately 30 interspersed ROs, inverters, phototransistors, condensers, and temperature sensors have been demonstrated as a proof of concept. All devices are able to operate individually and independently.

**Integrated transistor, inverter and ring oscillator on a microfibre.** Optical microscopic images and circuit diagrams of each electrical device are shown in Fig. 2A, D, respectively. A field-effect transistor (FET), a basic device element, in a top-gate and

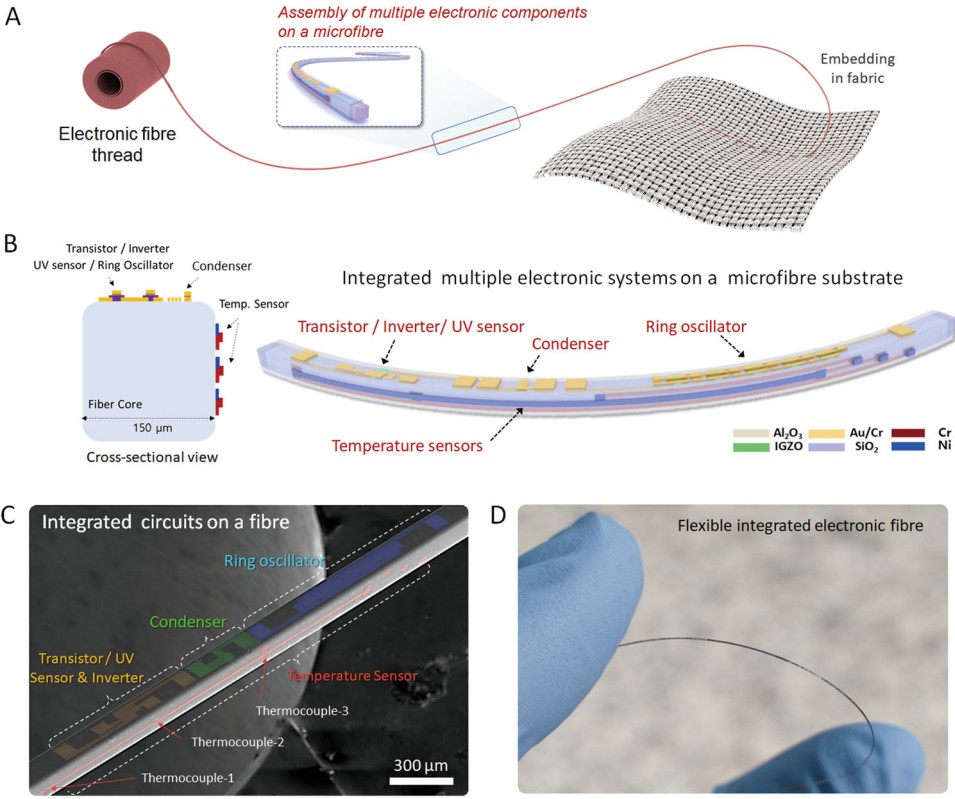

**Fig. 1 Assembly of multiple electronic systems on a microfibre. A** Schematic illustrations of e-textile integrated from a multifunctional electronic fibre and embedded in a fabric. **B** Cross-sectional and three-dimensional scheme of the device elements fabricated on the microfibre substrate. **C** SEM image of electronic devices fabricated on a single microfibre. **D** Photograph of the flexible electronic fibre.

bottom-contact (TG/BC) structure was fabricated to verify the capabilities of the miniaturized devices for electronic fibre applications. IGZO and $Al_2O_3$ are used as an amorphous oxide semiconductor and a gate dielectric, respectively. Their chemical composition was analysed with X-ray photoelectron spectroscopy (XPS), and the dielectric capacitance of the 15 nm thick $Al_2O_3$ layer was measured as $180\ nF\ cm^{-2}$, as described in Supplementary Fig. 4. Figure 2B shows the transfer characteristics of the driver FET in the depletion-load n-MOS inverter. The IGZO-based FET exhibits a field-effect mobility of $5.5\ cm^2V^{-1}s^{-1}$ in the saturation regime with negligible hysteresis and an on/off-current ratio greater than $10^7$ at a gate-source voltage ($V_G$) of 5 V and a low drain-source voltage ($V_D$) of 5 V. At the initial stage of fabrication, we got acceptable transfer curves of < 3 from the total of 8 transistors on 10 cm long proto-type electronic fibre, indicative of less than 40% of yield. Finally, we achieved almost 70% device yield by counting working transistors. Subsequently, five individual FETs were fabricated on each monofilament that exhibited an average saturation mobility of $5.5\ cm^2V^{-1}s^{-1}$ with a standard deviation of $1.1\ cm^2V^{-1}s^{-1}$, threshold voltage ($V_{Th}$) of $0.28 \pm 0.57$ V, and low subthreshold swing of $0.36 \pm 0.11\ V\ dec^{-1}$. These values are similar to those of previously reported IGZO-based FETs, indicating the validity of this fabrication process for integrated e-textile applications[22–25].

Based on the IGZO FETs, the electrical characteristics of both an inverter and a RO with 5 stages on the monofilament were evaluated, as shown in Fig. 2C, E, and F. Due to the complicated fabrication procedure and design of electronic circuits on fibre, we achieved only 60 and 40% of yield for inverter and RO, respectively. The depletion-load n-MOS inverter was implemented by a series connection between two n-MOS transistors, which play the role of driver and load. The source electrode of the load

transistor is connected to the gate electrode of the load transistor and drain electrode of the driver transistor. Channel widths ($W$) of 20 μm and 50 μm with the same channel length ($L$) of 10 μm for driver and load components were used, respectively, for the proper balance between driver and load transistors for the operation of the inverter and RO. The voltage transfer curve is measured for a bias voltage ($V_{bias}$) of 5 V and supply voltages ($V_{DD}$) of 2 V to 5 V. The output voltage-input voltage ($V_{Out}-V_{In}$) of the n-MOS depletion-load inverter is shown in Fig. 2C. Subsequently, the 5-stage RO was prepared by the depletion-load n-MOS inverter with IGZO channels as described above. The RO is connected in series with five depletion-load n-MOS inverters. The 1st inverter output becomes the 2nd inverter input, and the output of the 2nd inverter becomes the 3rd inverter input. This chain continues to the 5th inverter, and finally, the output of the last (5th) inverter returns to the input of the primary (1st) inverter (Fig. 2D). In this way, integrated circuits (ICs) were successfully fabricated using conventional semiconductor processes on a flexible fibre substrate. Although a higher process level and optimization for more refinement and accuracy are still required, it will be possible to integrate more complex ICs on the side of facets of rectangular fibre or a surface of a cylindrical filament. In addition, the output voltage waveform ($V_{out} - time$), oscillation frequency ($f$), and propagation delay ($\tau$) of the 5-stage RO according to the increase in $V_{DD}$ are described in Fig. 2E, F. The $\tau$ of the switching events was determined from fitting exponential functions to the measured $V_{Out}$ transitions that depend on $V_{DD}$. On increasing $V_{DD}$, $\tau$ increased and $f$ decreased.

**Optoelectrical characteristics of UV sensors on a microfibre.** To explore the possibility of multifunctional device integration on a

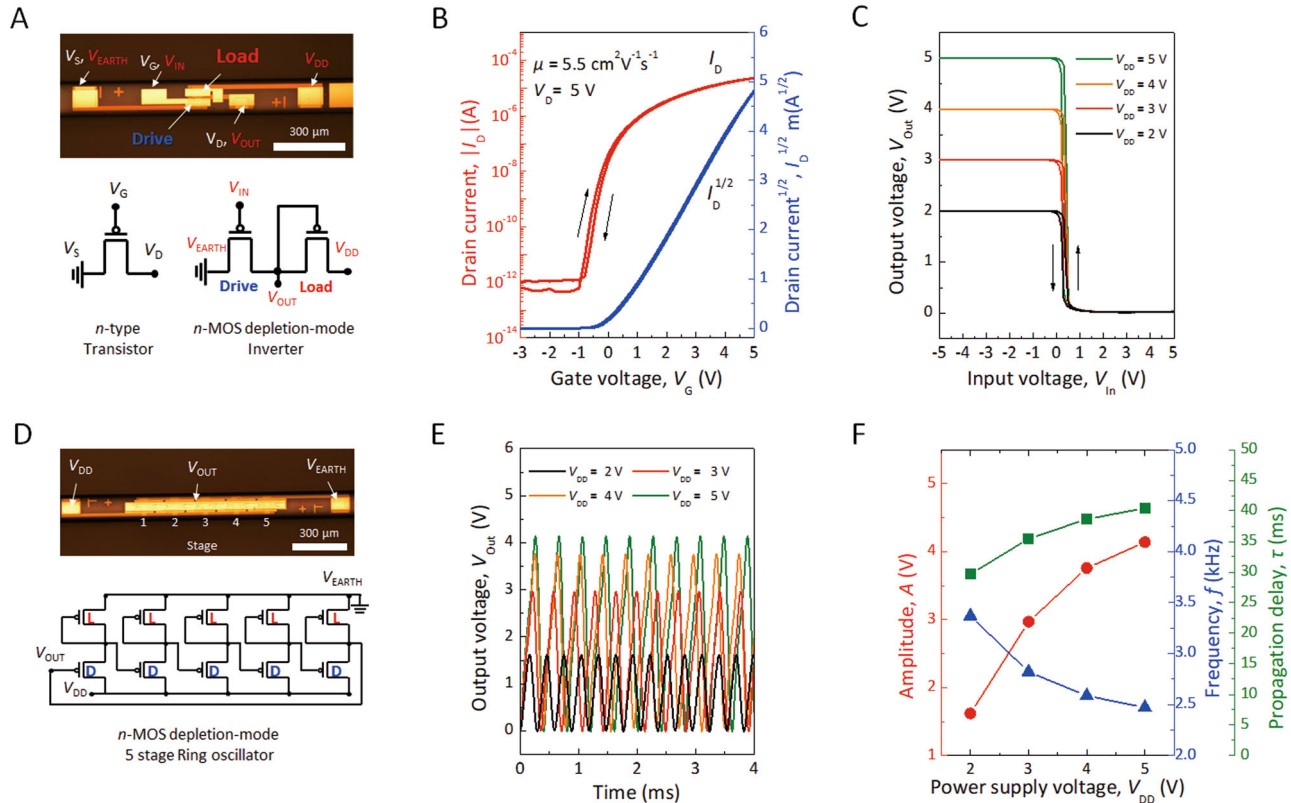

**Fig. 2 Integrated transistor, inverter and ring oscillator on a microfibre. A** photograph and circuit diagrams of an inverter based on IGZO-based FETs fabricated on a flexible fibre substrate. **B** $I_D-V_G$ curves of an IGZO FET representative of the fibre device. The FET has a channel length ($L$) of 10 μm and a channel width ($W$) of 50 μm. **C** Static transfer properties of the inverter based on two IGZO FETs for a supply voltage ($V_{DD}$) of 2 V ~ 5 V and bias voltage ($V_{bias}$) of 5 V. **D** Photograph and circuit diagram of a 5-stage ring oscillator based on depletion-load inverters fabricated on the microfibre. **E** Dynamic characteristics of the ring oscillator in response to different $V_{DD}$. **F** Oscillation amplitude, frequency, and propagation delay measured for $V_{DD}$ of 2, 3, 4, and 5 V.

fibre, we monitored the electrical signals of the sensor on a monofilament against changes in both UV light and temperature. UV light and temperature sensors were fabricated on two different sides of the optical fibre substrate. A UV sensing test was achieved by monitoring the optoelectrical characteristics of the single IGZO-based FET, enabling switching of the component. The IGZO semiconductor can be used for detection of the UV spectrum, as its optical bandgap is approximately 3.0 eV. Note that UV detection was carried out by measuring the change in drain current in the FET device. UV-LED light (wavelength: 470 nm) and UV-laser light (wavelength: 404 nm) irradiated both the top and bottom of the FET on a monofilament, implying UV sensing "out of fibre" and "through fibre core", respectively (Fig. 3A, D). Figure 3B presents the transfer characteristics of the IGZO-based FET on the fibre before and after UV exposure from out of fibre at $V_D = 5$ V with $V_G$ sweeping from $-5$ V to 5 V. Upon illumination, there is a significant increase in off-current from $4.0 \times 10^{-8}$ A to $7.5 \times 10^{-7}$ A at $V_G = 0$ V. This implies that exposed light contributes to the generation of photocarriers in the IGZO channel, inducing higher channel conductivity. Figure 3C displays the time-dependent photoresponse at different gate voltages of $-1$ V and 0 V with a drain voltage of 5 V under pulsed illumination by UV light (power intensity: 1.0 mW cm$^{-2}$). It should be noted that the irradiated UV light out of the fibre is partially blocked or scattered by the gate metal electrode due to the TG/BC structure of the FET device. Although the photo-to-dark current ratio is relatively low, it provides enough electrical signals that enable the detection of UV illumination under unknown environmental conditions (Fig. 3C). We also found one

more possible application as a signal transducer of the IGZO-based FET on the fibre. Figure 3D illustrates the schematics of the signal transducer. The UV laser was irradiated through the fibre core and propagated within the single FET fabricated on the optical glass fibre (Supplementary Fig. 5A). The off-state current in the $I_D-V_G$ curves remarkably increases by approximately three orders of magnitude when the IGZO semiconductor is excited by light propagation inside the optical fibre (Fig. 3E). The temporal response between the drain current and time ($I_D - time$) with various laser intensities showed stable switching and a relatively high photo-to-dark current ratio, while $V_D$ and $V_G$ were maintained at 5 V and $-5$ V, respectively (Fig. 3F). It will be possible to realize a high-performance photosensor or signal transducer by using photosensitive semiconducting materials and different device architectures, such as bottom-gate/top-contact device architectures and perpendicular diodes. The photocurrents of the electronic fibre were extracted from the transfer curves at $V_G = -5$ V and $V_D = 0$ V, and the photoresponsivity ($R_\lambda$) was calculated by

$$R_\lambda = \frac{I_{light} - I_{dark}}{P_{opt} \times A} \tag{1}$$

where $I_{light}$ and $I_{dark}$ are the drain current under light illumination and dark conditions, respectively; $P_{opt}$ and $A$ represent the incident illumination power (1.0 mW cm$^{-2}$ for 470 nm LED light, $P_{opt} = 84.5$ μW cm$^{-2}$ for 404 nm laser light, the illumination power was measured by a power metre) and effective area, respectively. $A$ is the channel area (width × length = $2 \times 10^{-6}$ cm$^2$)

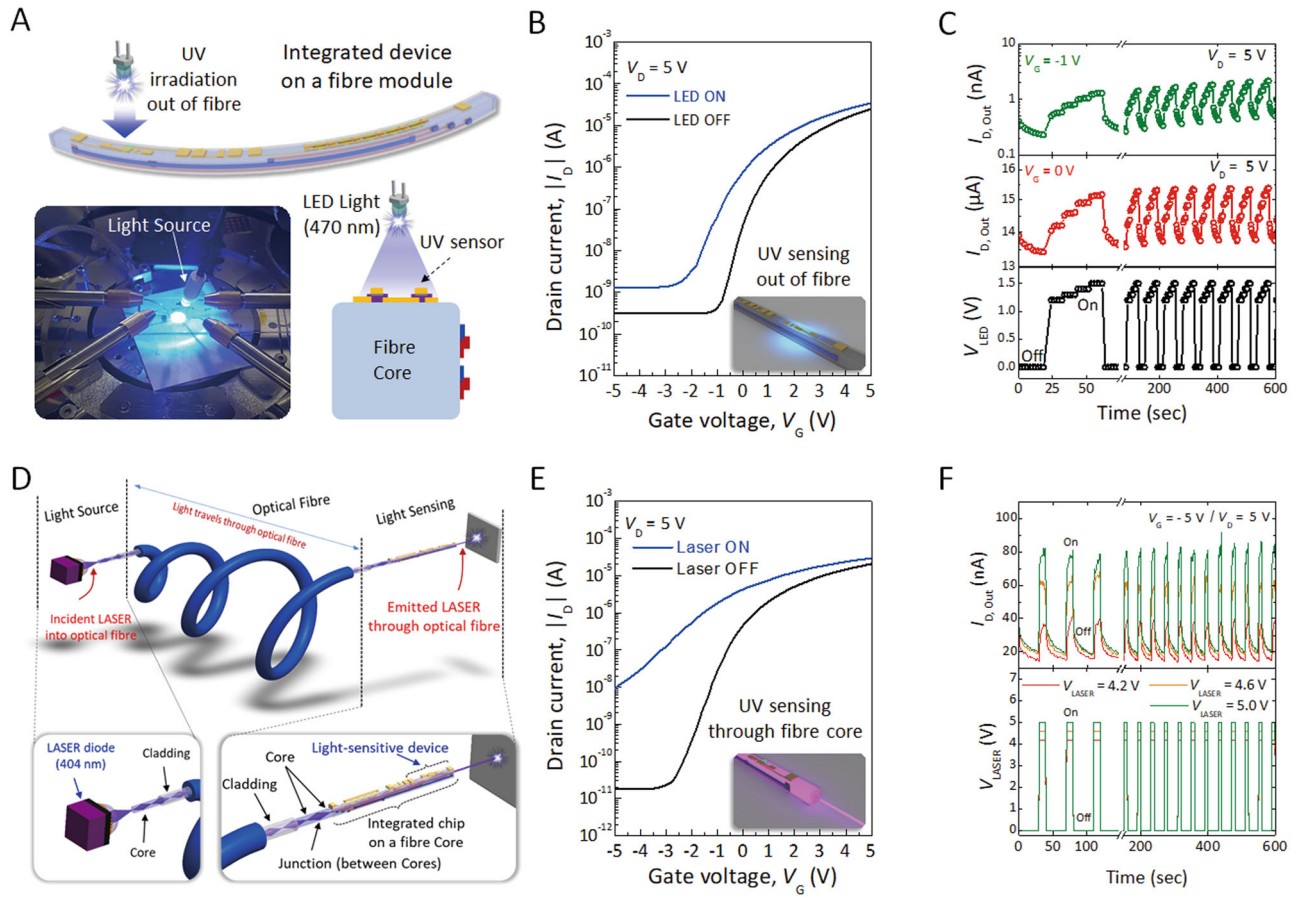

**Fig. 3 Optoelectrical characteristics of UV sensors on a microfibre. A** Schematics and a photograph of the optoelectrical measurement when outside of the fibre device is irradiated by 470 nm UV-LED light. **B** Transfer curves of the IGZO-based phototransistor in the dark and under UV light exposure. **C** Time-dependent photoresponse at different gate voltages under pulsed illumination with UV light. **D** Schematics of the optoelectrical measurement when the inside of the fibre device is irradiated by 404 nm laser light. **E** Transfer characteristics of the phototransistor in the dark and during exposure to UV light in the fibre core. **F** Transient photocurrent of the IGZO device with different laser powers of 4.2 V, 4.6 V, and 5.0 V.

of the device. The phototransistors presented a photoresponsivity of 0.64 A W$^{-1}$ at UV-LED light irradiated out of the fibre and 53.9 A W$^{-1}$ under UV laser light propagated through the fibre core, as shown in Fig. 3B, E. In this regard, IC on optical fibres can be utilized not only as a photodetector but also to construct wireless sensor networks that are powered by laser beam propagation[26].

**Thermoelectrical characteristics of temperature sensors on a microfibre.** Resistive-type sensors are directly integrated on the other side of the fibre to allow multifunctionality of the chip on a fibre, as shown in Fig. 4. For efficient measurement and detection of thermal information, Ni and Cr were selected as thermoresistive materials because these pure metals can be easily deposited by vacuum thermal evaporation and have high Seebeck coefficients ($-19$ μV K$^{-1}$ for Ni and $+20$ μV K$^{-1}$ for Cr), which can generate large thermoelectric voltages and signals for temperature monitoring (Supplementary Fig. 6)[27]. The interval distance between each thermocouple is 3.4 mm, and the contact pads of the three thermocouples are located on one side of the fibre surface. The temperature sensors on a monofilament operate through voltage changes induced in response to the temperature at different positions along the fibre. These multiple integrations of sensors on the fibre enable precise monitoring of temperature under environmental conditions. By setting up the circuit and sharing the ground contact, the temperature can be measured at three points simultaneously (Fig. 4A and Supplementary Fig. 5B).

Furthermore, the change in thermoelectric voltages ($\Delta V_{TE}$) with increasing temperature of the thermal source ($T_{Source}$) and with the temperature difference between thermally synchronized thermocouples was measured at a given temperature and room temperature ($T_{TC} - T_{RT}$). A detailed discussion of each sensor is described in Supplementary Fig. 6.

Due to the unique shape of our chip on a fibre, it can be applied as an implantable temperature sensing module, as shown in Fig. 4B. To monitor the temperature of the heat source, the integrated sensing fibre tip is carefully implanted into a hot block. As a result of thermal conduction from the thermal source to the sensor through the body of fibre, the temperature in a material was successfully monitored spontaneously by changing the temperature of the heat source. The thermoelectric voltages ($\Delta V_{TE}$) of each thermocouple on a monofilament linearly responded by changing the temperature of the heat block from room temperature to 60 °C, exhibiting lower values in order away from the thermal source ($T_{Source} > T_{TC-1} > T_{TC-2} > T_{TC-3}$) (Fig. 4C). Although the detected temperature decreased exponentially as the position of the temperature sensor moved away from the heat source due to heat loss from air convection, as shown in Fig. 4D, the calculated temperature at each integrated sensor on a monofilament exhibited clear stepwise behaviour. This implies that the integrated 1D thermoresistive sensors are applicable to not only wearable temperature sensing network systems but also implantable modules. Hence, the above results, together with the UV/thermal sensing and electronic components on the fibre, can

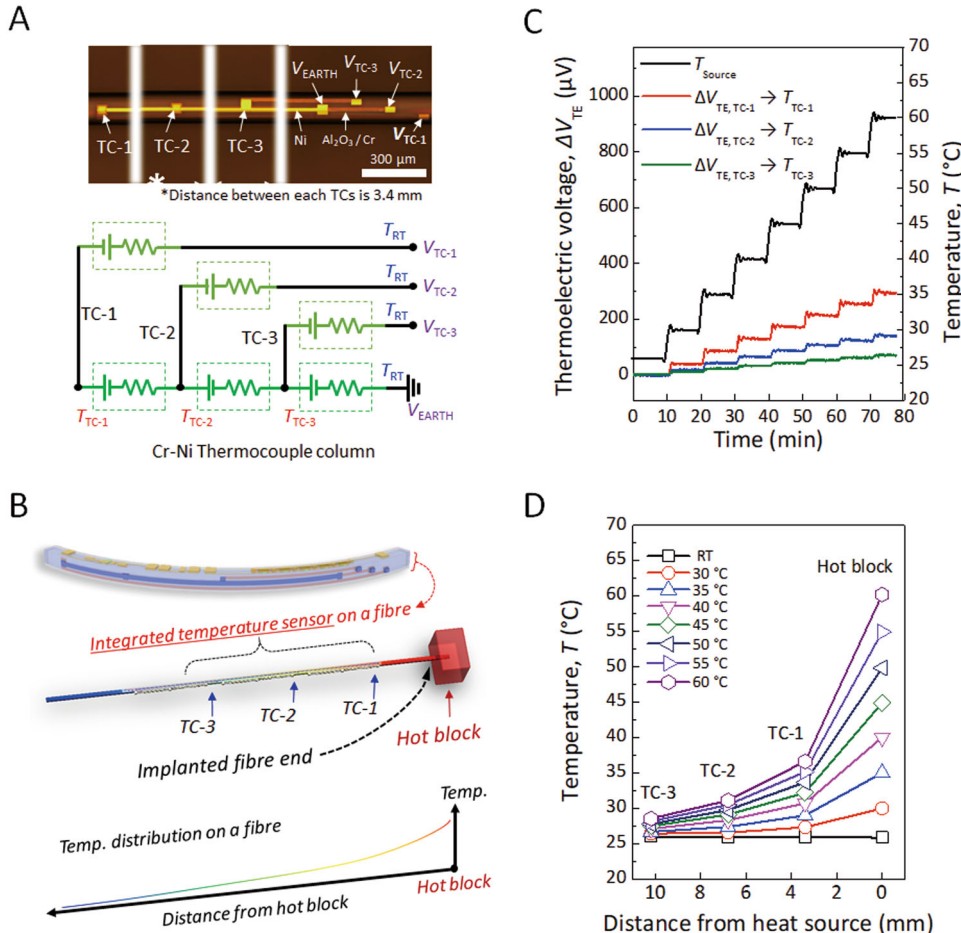

**Fig. 4 Thermoelectrical characteristics of temperature sensors on a microfibre. A** Photograph and schematic diagram of the thermosensing components. **B** Schematic illustration of the integrated thermocouples on a fibre and temperature gradient across the fibre device. **C** Changes in thermoelectric voltages of each thermocouple as a function of temperature. **D** Temperature distribution as a function of distance from the heat source to each sensor.

offer substantial promise for the implementation of high-performance and multifunctional electronic fibre systems for future electronic textile applications.

**Integrated electronic fibre under various bending conditions and embedding in a fabric**. To test the flexibility and stability of the multiple electronic systems on a monofilament, the IGZO FET device on the fibre was measured under both tensile and compressive stress conditions, as shown in Fig. 5A, B. For systematic analysis of the IGZO FET on the fibre under various stress conditions, we prepared two electronic fibres that were fabricated in different batches. The fibres were carefully placed and fixed on flexible polyethylene terephthalate (PET) substrates by using polyimide tape for both concave and convex bending conditions. The engineering strain of the fibre ($\varepsilon$), which is presented as the ratio of the total deformation to the initial state under applied mechanical input, is calculated through the equation below:

$$\varepsilon = \frac{d_s + d_f}{2R} \frac{(1 + 2\eta + \chi\eta^2)}{(1 + \eta)(1 + \chi\eta)} \approx \frac{d_s + d_f}{2R}, \quad (2)$$

where $\eta = d_f/d_s$ and $\chi = Y_f/Y_s$. $d_s$ and $d_f$ are the thicknesses of the substrate (square shaped-glass fibre, 150 μm) and active layer (IGZO, 15 nm), respectively. $Y_s$ (glass, 50−90 GPa) and $Y_f$ (oxide, >100 GPa) are the Young's modulus of the substrate and active layer, respectively. $R$ is the bending radius. In the simplified

formula under the premise that the substrate is much thicker than the active layer and there is a relatively small difference between $Y_s$ and $Y_f$, $\varepsilon$ within the active layer on the bent substrate can be roughly obtained as the right term of the equation[28].

The field-effect mobility ($\mu_{sat}$), threshold voltage ($V_{th}$) and drain current ($I_{D,on}$) in the on state are estimated from Fig. 5A–C. Due to the device-to-device uniformity, we observed slight differences in $I_D − V_G$ characteristics between two electronic fibres for concave and convex bending. However, the electrical parameters, such as field-effect mobility, threshold voltage, and drain current, of each IGZO FET on both fibres maintained their initial switching performances up to a compressive strain of 0.64% and a tensile strain of 0.68%, respectively. To examine the mechanical durability of the fibre, a repeated bending cycle test was carried out, as shown in Fig. 5D. The IGZO FET on the fibre sustained 10,000 cycles of repeated bending at a bending radius of 11.7 mm. We did not observe apparent breakdown of the fibre or delamination of the semiconductor or metal electrode during or after the bending test. The saturated mobility and threshold voltage of the FET slightly decreased from 3.77 to 3.73 cm$^2$V$^{-1}$s$^{-1}$ and −0.75 V to −0.81 V, respectively. The drain current of the FET was measured to be ~1.28 μA at each bending condition with negligible changes, as shown in Supplementary Fig. 7. Although the electronic fibre was exposed to air without any passivation layer during the repeated bending and $I−V$ measurement, the IGZO FET exhibited electrical stability and endurance without serious degradation and malfunctions. From the additional mechanical tests and its electrical

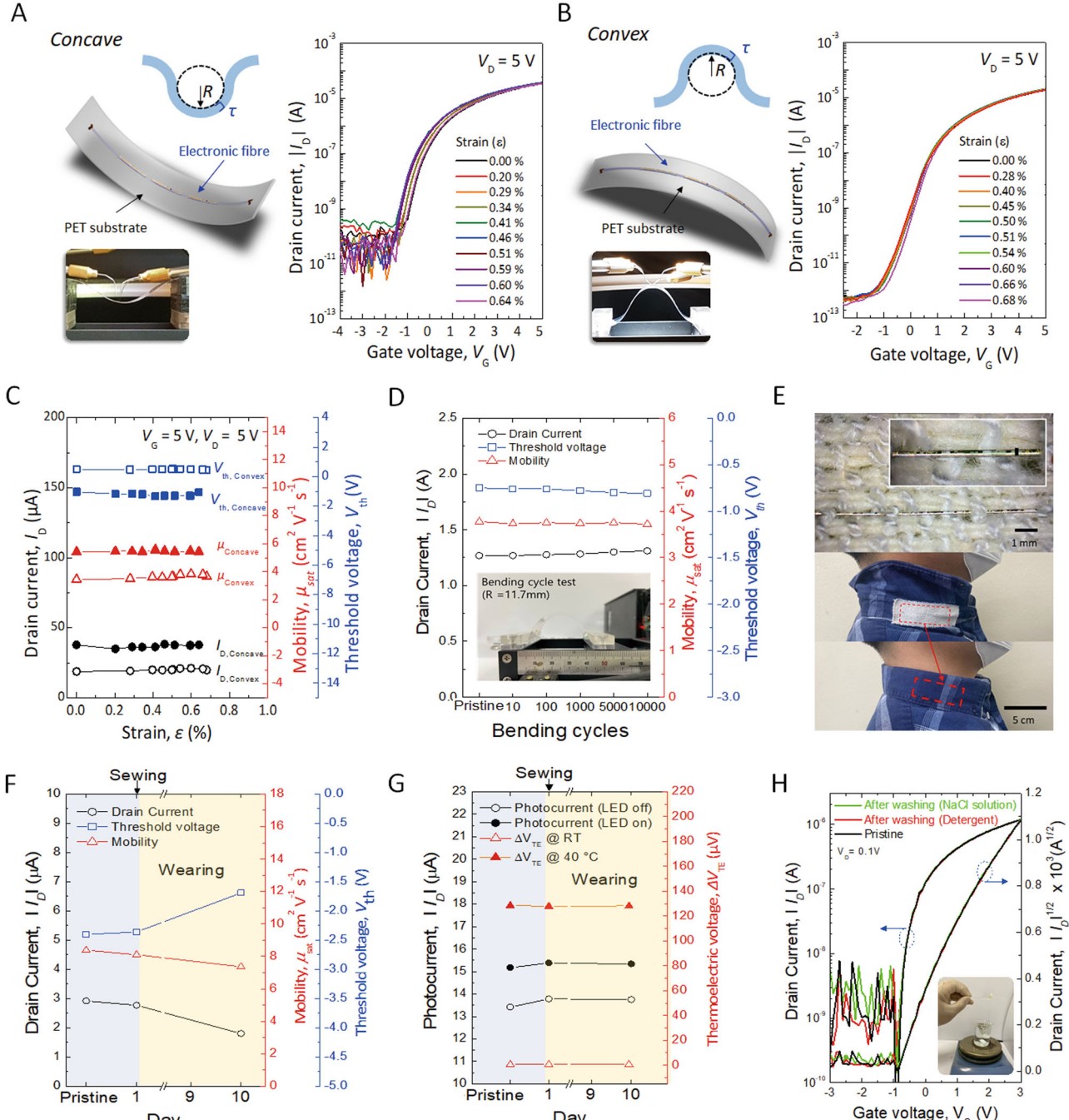

**Fig. 5 Integrated electronic fibre under various bending conditions and embedding in a fabric.** Schematic illustrations of the integrated fibre device on PET substrate (left) and $I_D$–$V_G$ characteristics (right) of the IGZO FETs under **A** concave and **B** convex bending conditions. The mechanical strain ($\varepsilon$) is calculated using the bending radius ($R$). The insets show photographs of the flexible fibre device measured during bending. Change of on-current states, field-effect mobilities, and threshold voltages of the IGZO FETs on the fibre **C** as a function of mechanical strain with forwards and backwards bending, and **D** during repeated bending test up to 10,000 cycles at the bending radius of 11.7 mm. **E** Photograph of the electronic fibre embedded in a fabric and sewed inside the collar of a shirt. **F** On-current states, field-effect mobilities, and threshold voltages of the IGZO FETs on the fibre at pristine, sewed, and after wearing for 10 days. **G** Photocurrent of the photosensor and thermoelectric voltage of TCs on the fibre at pristine, sewed, and after wearing for 10 days. **H** Electrical characteristics of the encapsulated IGZO FET on the electronic fibre before and after washing (detergent and NaCl solution).

characteristics of the electronic fibre to the limit, we found that the threshold bending radius ($R$) and strain ($\varepsilon$) of the electronic fibre were 7.3 mm and 1.03%, respectively, without the mechanical failure of the silica fibre. (Supplementary Fig. 8) Because the electronic performance of the IGZO TFT on the fibre were maintained right before the breakdown of the fibre, the mechanical durability of our electronic fibre seems to depend on the

mechanical properties of the fibre substrate. Due to the limitation of the mechanical properties of inorganic material-based microfibres, it was difficult to achieve better flexibility than that of conventional flexible electronic devices. However, it is essential to use high-performance inorganic semiconducting materials to implement high-performance and integrated electronic fibre system. Especially, silica-based microscale fibre can be considered

as one of the promising fibre substrates due to its capability of achieving better electronic properties and scaling down by introducing both high-temperature and conventional semiconductor fabrication processes. Nevertheless, it is still required to develop better fibre substrate that enables both high performance and flexibility integrated electronic fibre system. For example, flexible polymeric materials, such as polyimide, include a combination of excellent properties, such as chemical stability, thermal stability, low thermal conductivity, radiation resistance, insulativity, high tensile strength and tensile modulus, which can be considered as substitutional fibre substrates for better flexibility and processability[29].

Wearable e-textiles should be breathable and washable with high flexibility and shape adaptability. To demonstrate feasibility for a potential e-textile application, the electronic fibre was directly sewed in a piece of common compression bandage. To prevent unexpected damage to the electronic fibre during the sewing process, we introduced a special sewing method by using a needle of a syringe that enables the fibres to place in the fabric safely (Supplementary Mov. 1). It helped us to sew the electronic fibre in the fabric without any serious damage to the surface. Because the syringe needle guides the electronic fibre during sewing, we were able to control the orientation of the fibre in the garment(bandage), resulting in the UV sensor or temperature sensor upward for both electrical contacts for UV or temperature sensing. Then, the fabric was sewn inside of the collar of a shirt again without any additional protective coating, as shown in Fig. 5E. We monitored the switching performance of the IGZO FET, optoelectrical characteristics of the UV sensor, and thermoelectrical characteristics of the temperature sensor on the fibre at pristine, sewed and after wearing for 10 days (Supplementary Fig. 9). The IGZO FET maintained its electrical performance for 1 day. After 10 days, it exhibited a slightly degraded field-effect mobility approximately $7.4\,cm^2V^{-1}s^{-1}$ (Fig. 5F). Additionally, we observed a slight decrease in drain current from 2.9 to 1.8 µA and a positive shift in threshold voltage from −2.4 to −1.7 V, respectively. From the additional measurements of the photocurrent and thermoelectric voltage of the electronic fibre before and after wearing for 10 days, as shown in Fig. 5G, we found that the photosensor was relatively stable, maintaining photocurrent values of 15.3 and 13.7 µA for the LED on and off states, respectively, at $V_G = 0\,V$ and $V_D = 5\,V$. Although our electronic fibre in a garment successfully detected UV signal out of fibre, it is due to the syringe needle-assisted sewing method. If the electronic fibre is completely buried in a garment by the conventional sewing method, it is hard to expect proper detection of UV light illuminated from all directions in general. Therefore, further research on how to place electronic fibre in a garment and evaluate its sensing performances is still required for practical UV sensing electronic fibre applications. Additionally, we evaluated the thermoelectrical characteristics of each TC on the fibre in the fabric placed on a hot chuck. All TCs exhibited almost the same value of approximately 0.52 µV of thermoelectric voltage at room temperature (23 °C) before and after wearing for 10 days. Similarly, the average thermoelectric voltage of each TC (TC1, TC2, TC3) showed a similar value of approximately 128.3 µV at 40 °C before and after wearing for 10 days. Although our electronic fibre worked well above stressful conditions, we should note here that an additional thick protecting or shielding layer is required to eliminate expected risks related to their contact with the human body while keeping the electrical function of the devices against various environmental conditions (mechanical stress, chemicals, sweat, etc.)[30,31]. To further evaluate the washability of the electronic fibre, we carried out the CTAC process (speed $1.0\,mm\,min^{-1}$) of SU-8 solution to form a passivation layer (thickness = 2 µm). The

electronic fibre was completely covered by an SU-8 passivation layer. Then, the encapsulated electronic fibre was dipped in detergent solution (5 ml in 90 ml of tap water) and NaCl solution (0.5 wt% for artificial human sweat) for 30 min and rinsed in pure tap water with stirring at 600 rpm at room temperature[32]. After washing, the electronic fibre was dried at 60 °C on a hotplate. Because the SU-8 layer completely covered the outer electronic fibre, the electronic performance of the IGZO FET showed a negligible difference in transfer characteristics (field-effect mobility of $3.74\,cm^2V^{-1}s^{-1}$ in the saturation regime and an on/off-current ratio of 4 orders of magnitude) before and after washing with detergent and 0.5 wt% NaCl solutions, as shown in Fig. 5H. This implies that the encapsulated electronic fibre maintained a stable performance, regardless of the wet environment, such as washing and perspiration conditions. It may be possible to implement practical electronic fibres by introducing a reliable protecting or encapsulation layer that is durable under various mechanical or chemical conditions. Therefore, we believe that our electronic fibre platform is still considered a valid approach for integrated electronic textile systems.

## Discussion

In summary, we demonstrated an electronic fibre platform with integrated electronic devices on a 1D microfibre. Our electronic fibre system was composed of basic electronic units such as transistors, inverters, ring oscillators for data processing, and sensing or transducing units for detecting optical/thermal signals. For high integration density, the capillary-assisted coating method and maskless photolithography were implemented to quickly and directly draw the desired device design at high resolution under ambient conditions. Due to the limitation of scaling down in the laboratory-scale fabrication process at the current status, we achieved 30 device sets (e.g. transistor with $L = 10\,µm$ and $W = 50\,µm$/Contact pads with $100\,µm \times 100\,µm$, $50\,µm \times 50\,µm$) on a 10 cm long fibre, demonstrating as a proof of concept to implement direct assembly of electronic systems on a microfibre. If the semiconductor fabrication technology on a microfiber substrate is matured, we believe that it is possible to implement a higher density electronic fibre similar to those of conventional semiconductor devices on a silicon wafer. For more information on the feasibility of our electronic fibre platform, we calculated simply how long fibre requires to integrate a personal computer microprocessor (Intel Pentium processor) which has a die area of $91\,mm^2$ and contains 3.3 million transistors with a process step of 0.35 µm BiCMOS technology (https://www.intel.com/pressroom/kits/quickreffam.htm#pentium). If the same fabrication technology is applied to integrate the above chip on the outer shell of the circular microfibre with a diameter of 150 µm, it requires only 19.3 cm long microfiber to implement microprocessor fibre, as shown in Fig. 6. Therefore, the proposed device platform provides an architecting type of fibrous device and believe to contribute to the realization of high-density electronic fibres embedded in clothes. We envision that this assembly of multiple electronic systems on microfibres will enable technological advances in electronic textiles as well as conventional batch-process-based 2D wafer electronics by adapting a reel-to-reel continuous fabrication process. Meanwhile, to implement integrated electronic fibre by reel-to-reel process, precise and continuous control of the orientation of the fibre face is necessary to enable a continuous fabrication process during the feeding of the fibre. To overcome size and density limitation of integrated electronic components on a fibre for real applications, it is required to introduce a higher resolution of maskless photolithography process and modify the location of exposure modules enabling to use the entire circumference of fibre. Finally, the

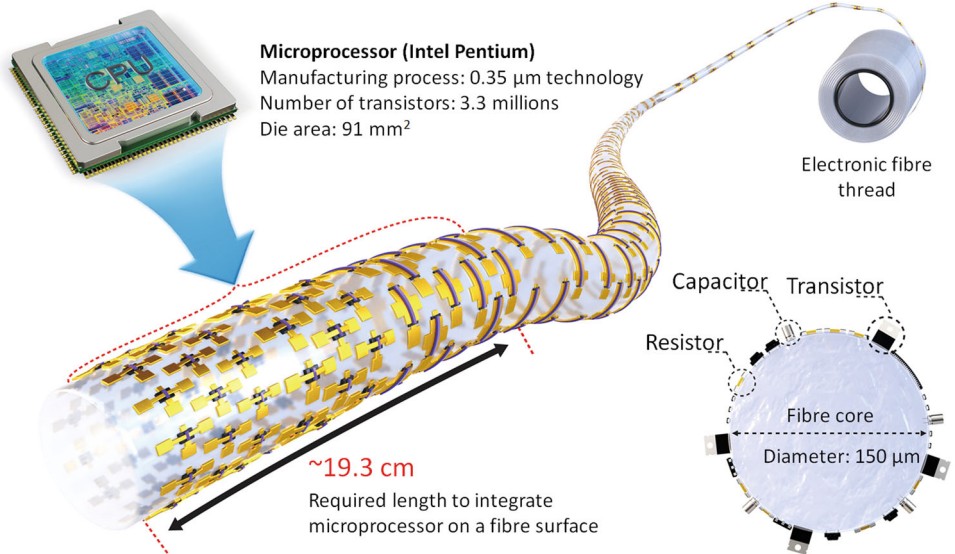

**Microprocessor (Intel Pentium)**
Manufacturing process: 0.35 μm technology
Number of transistors: 3.3 millions
Die area: 91 mm²

Electronic fibre thread

Capacitor   Transistor

Resistor

Fibre core
Diameter: 150 μm

~19.3 cm
Required length to integrate microprocessor on a fibre surface

**Fig. 6 Schematics of the implementation of the microprocessor on a microfibre.** Theoretical calculation on how long microfiber (diameter of 150 μm) requires to integrate the Pentium microprocessor (https://www.intel.com/pressroom/kits/quickreffam.htm#pentium).

optimization of the arrangement of the equipment for each process step (e.g. coating, lithography, development, deposition, etching, and inspection) is also considered as an important factor to perform mass production of electronic fibre by continuous reel-to-reel process. (Supplementary Fig. 10).

## Methods

**Materials preparation**. All materials used in this study were purchased as follows without any purification. Optical fibre with square-shaped silica core (FP150QMT, Thorlabs), gold (Au, 99.99%, TAEWON SCIENTIFIC), chromium (Cr, CR-090010, 99.9%, Nilaco), nickel (Ni, NI-311165, 99.9%, Nilaco), IGZO sputtering target (In$_2$O$_3$: Ga$_2$O$_3$: ZnO = 1: 1: 1 in atom%, 99.99%, Advanced Engineering Materials), Au etchant (Gold Etchant, Sigma Aldrich), Cr etchant (CR-7, KMG Electronic Chemicals), Ni etchant (Nickel Etchants, TRANSENE), Al$_2$O$_3$ etchant (Aluminium Etch ANPE 80/5/5/10 Microchem), IGZO etchant (HCl, 35%, Wako), Positive photoresist (AZ GXR 601, AZ Electronic Materials), Developer (AZ 300 MIF Developer, Merck).

**Integration of electronic components on a microfibre**. The square-shaped silica core microfibres (150 μm × 150 μm × 7.5 cm) were ultrasonically cleaned in deionized water, acetone, and isopropanol for 5 min. This was followed by ultraviolet-ozone (UV/O$_3$) treatment for 15 min. Metal electrodes were patterned through maskless photolithography. A 2 μm thick photoresist layer was coated on the fibre substrate using a capillary tube-assisted coating (CTAC) process (speed 1.0 mm min$^{-1}$), baked on a hot plate at 100 °C for 2 min, and exposed to ultraviolet light using a maskless aligner (MLA 100, HEIDELBERG) with an energy density of approximately 200 mJ cm$^{-2}$ and 1000 μm s$^{-1}$ of driving speed. The sample was immersed in developer for 2 min and rinsed with deionized water after hard baking (100 °C for 2 min). A 10 nm thick Cr adhesion layer, followed by 30 nm thick Au, was deposited and patterned by soaking the fibre substrate in a bath of resist remover. Notably, 50 nm thick Cr and 50 nm thick Ni layers were deposited by vacuum evaporation at a base pressure of ca. ~10$^{-6}$ torr and speed 0.5 Å s$^{-1}$ for thermocouples. IGZO thin films (15 nm) were deposited using an AC sputter (ACT ORION 8 Sputtering System, AJA International, 100 W, Ar: O$_2$ = 20.0: 0.2 sccm, 2 × 10$^{-3}$ torr). After deposition, the as-deposited IGZO films were placed on a hot plate and thermally annealed for 30 min at 300 °C in ambient air to improve the quality of the IGZO film. Al$_2$O$_3$ (thickness of 36.1 nm) for the gate dielectric and encapsulation layers was directly deposited by the atomic layer deposition system (LUCIDA D100 ALD, NCD). Trimethylaluminum and deionized water were used as the precursors and oxidants in this system, respectively. The substrate temperature was maintained at 100 °C during the 400 cycles of the deposition process. In the wet-etching sequence, each etchant for each material was purchased commercially and used after being diluted with deionized water. The detailed conditions are as follows. Au etchant: 1/20 for 3 min; Cr etchant: 1/20 for 3 min; Ni etchant: 1/20 for 3 min; IGZO etchant: 1/100 for 2 min; Al$_2$O$_3$ etchant: 50 °C for 4 min. After each wet-etching process was completed, the samples were washed with deionized water, transferred to an acetone bath at 100 °C, and immersed for 5 min to remove the photoresist.

**Device and thin film characterization**. Current-voltage characteristics were measured with a HP4145B (HP Ltd.), Keithley 4200SCS (Keithley Instruments, Ltd.), and a digital phosphor oscilloscope DPO2002B (Tektronix, Ltd.) in ambient air. The optoelectrical and electrical characteristics of the phototransistor were measured using a Keithley 4200 semiconductor characterization system under illumination at a wavelength of 470 nm (UV-LED light of 50 mW cm$^{-2}$). To measure the optoelectrical signal of UV light travelling through the fibre, 404 nm UV laser light (50 mW) was employed. To sew electronic fibre in a fabric, a special sewing method was developed by using a needle of a syringe that enables the fibres to place in the fabric safely (Supplementary Movie 1). The coating of the encapsulation layer (SU-8) on the electronic fibre was processed by using the CTAC process (speed 1.0 mm min$^{-1}$). To evaluate the washability of the electronic fibre, detergent (5 ml in 90 ml of tap water) and NaCl (0.5 wt% for artificial human sweat) were well deserved in tap water. The washing test was performed for 30 min and rinsed in pure tap water with stirring at 600 rpm at room temperature. After washing, the electronic fibre was dried at 60 °C on a hotplate. SEM and optical microscope images were obtained using a Nova NanoSEM 450 (FEI Ltd.) and Nikon ECLIPSE LV150 microscope (Nikon), respectively. The thickness of thin films was determined from a surface profiler (ET200, Kosaka Laboratory Ltd.). X-ray photoelectron spectroscopy (XPS) measurements were performed using an ESCALAB250Xi (Thermo Fisher Scientific, USA) at a basic pressure of 10$^{-9}$ mbar.

## Data availability

The imaging data that support the findings of this study are available from the corresponding author upon reasonable request. Source data are provided with this paper.

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

## Acknowledgements

This work was supported by the National Research Foundation of Korea (2020R1A2C2010163).

## Author contributions

T.-W.K. developed the idea. S.H., A.L., T.-W.K. conducted the experiments, and S.H., M.K., A.L., S.B., S.-K. L., S.H.L., T.L., G.W., T.-W.K. collected and analysed the data. S.H., M.K., T.-W.K. wrote the manuscript. All authors discussed the results and commented on the manuscript. T.-W. K. oversaw the project, revised the manuscript, and led the effort to completion.

## Competing interests

The authors declare no competing interests.
