## [Peer Review File · Nature Communications]

Integration of multiple electronic components on a microfibre towards an emerging electronic textile platformEditorial Note: Parts of this Peer Review File have been redacted as indicated to remove third-party material where no permission to publish could be obtained.

REVIEWER COMMENTS

Reviewer #1 (Remarks to the Author):

The English language of the paper is not vetted.

The notable result is to provide a fibre that integrates multiple components. This opens up prospects for future textile-electronic platforms.

The proposed approach is interesting even if the final concept has properties that are a little different from the textile. The support remains rigid. This point will be improved for future generations. The approach is complementary to works like <https://doi.org/10.1038/s41467-021-23628-5>

The work presented appears rigorous and serious and is well presented. However, I have a few questions and remarks (minor revision):

Throughout the paper, avoid carriage returns between numbers and "fig.". (Line 111-112, 117-118, etc.)

The summary is well constructed and summarizes the main lines of the article in a clear and precise manner.

Line 86: I don't understand the part of the sentence "...the cuboid shape silica fibre includes six planar faces in 3-axis...". The microfiber is square-shaped, therefore it has 4 planar faces. The ends are not exploited for the components (?).

Line 157: why a hyphen between "470" and "nm" (and "404" and "nm")?

Line 177, 178: replace "A/W" with "A W-1"

Line 227: 0,64 and 0,68 % of strain is very low for textile fibre (as you write line 232). Why haven't you tested more? Do you know the limit, or the mechanical behaviour, of your microfiber substrate? It would have been interesting to carry out the mechanical tests to the limits and to make the electrical measurements at the same time.

Line 248: For the perspectives, in addition to the change of support to work with a more flexible microfiber substrate, is it possible to work with a circular section fibre and to better exploit the entire circumference to increase the density of electronics components?

Reviewer #2 (Remarks to the Author):

Review: Integration of multiple electronic components on a microfiber toward emerging electronic textile platform

Overall, this is a very interesting paper with a potential platform fabrication process which would benefit e-textile development. I would recommend publication but with some changes as listed below.

This paper would benefit from a native English speaker to improve the grammar, some of the sentences are clunky and make reading, what is otherwise an interesting paper, difficult at times. Particular focus should be given to the appropriate use of 'the' which is often missing. Despite this minor comment, it is well written overall and the principle discussion is easy to follow.

I believe the noteworthy result is the ability to fabricate basic circuits on the very fine fiber/yarn which could enable complex circuitry to be embedded into e-textiles.

The literature review is good but would benefit with the inclusion of some more contemporary e-

textile yarn comparisons, for example the work at Imperial College London, Nottingham Trent Advanced Textile Research Group, University of Southampton and University of Glasgow. Their work is competitive with existing electronic components embedded into e-yarns or filaments; I believe this paper is taking this concept further with circuits fabricated on the yarn itself which is impressive.

I have some questions on the fabrication methodology which appears sound but it would be good to have more detail on how this could be scaled up and the current size limitations but these are contained within my points below.

I have some specific comments:

Line 54 – please explain what you mean by “any patterning on as-processed” do you mean there is no patterning process during manufacture?

Line 87 – what temperature range can it sustain? “high temperature” should be defined – e.g. 90 °C is a high temperature for fabrics, but a low temperature for most electronic fabrication methods.

Line 139 – do you have any information on the yield of these circuits; I realise these are prototypes so any yield should be taken in that context but it would be good to know how difficult this process is to achieve.

Line 143 – Is there any difference in device quality when fabricating on the rectangular or cylindrical fibre?

Line 151 – Did you perform any measurements of the UV sensor response once it is embedded in the textile? Please comment on the potential issues, if any, that you envisage – for example you would be unable to control the orientation of the fiber in the garment.

Line 158 – Linked with the above, for the out of fiber and through core measurement, you would not be able to control the direction once embedded in a garment, so can you separate these two measurements out at all to therefore determine the intensity of UV regardless of direction?

Line 161 – this is unclear “was responded against”

Line 183 – Some excellent examples of potential applications and a platform technology for more complex circuit/sensors in the future. However, it is not clear how big the circuits could be in terms of continuous length along the fiber, e.g. not just a string of individual circuits but a single interconnected circuit?

Line 207 – How would the readings be affected by being inside a fabric? In this scenario the heat block would not be at the end but would be in parallel to the thermocouples, do you envisage any problem or would they all just give the same reading in such a close area?

Line 208 – would it be better for the e-textile to just have one large sensor rather than 3 individuals?

Line 229 – unclear text – “was kept its device parameters”

Line 235 – you say conventional semiconductor fabrication process – please give more detail on any modifications required or size limits when using this approach with your particular system and future potential scale up in both production volume and device/system size.

Line 246 – please elaborate on ‘plastic microfiber’ alternatives; you previously said you were using fused silicon because it allowed the use of inorganic material deposition, so which material are you suggesting and would this affect this requirement?

Additional general comments:

1. It's not entirely clear which is the main novelty being claimed in the paper – is it the ability to process these devices on the fiber, is it the process itself or the idea of using it for e-textiles? It does feel as if the e-textile part has just been tacked on because the integration of this is only mentioned in Fig1A and doesn't appear to be tested beyond the concept? E-textile is in the title so more should be added to discuss the requirements for this and any tests or knowledge that has been obtained to identify these fibers as suitable.

2. It is more conventional in textile engineering to define the substrate as a yarn in your case, fiber would refer to all the smaller fibers that are then spun together to form the yarn. I would recommend either changing it throughout or might be easier to put a caveat at the beginning to explain your choice of terminology, e.g. "in this paper we define a 'fiber' as..." and relate that to conventional textile terminology.

3. Fused silica is a relatively stiff ceramic compared to traditional textile or e-textile materials; how flexible is any fabric containing these microfiber yarns?

4. There is no detail on how these devices are connected to, either for testing or for use in a garment; has this been considered, please comment even if it is future work.

5. Please comment more on the potential density of devices, limits to things such as track width, transistor feature sizes, even if they are just theory it would be good to know to see the potential for this methodology.

Reviewer #3 (Remarks to the Author):

Comments to authors:

In the study, the capillary-assisted coating method and the maskless photolithography were implemented onto rectangular optical fiber (non-planar transparent substrate, a square-shaped microfiber made of fused silica) to fabricate quickly miniaturized functional electronic devices such as transistors, inverters, ring oscillators for data processing, as well as sensing or transducing units for detecting optical/thermal signals. Although the etching of optical fibers for sensing applications is not novel idea and studied a lot in the literature, the suggested assembly of multiple electronic system on the optical microfiber would enable new technological advances if the below most common major problems and challenges from wearability concept and textile manufacturing view point are solved and discussed for the emerging field of e-textiles in the paper:

-Optical fibers are not commonly preferred in textile and clothing manufacturing techniques due to the limitation of their mechanical properties. In the study, inorganic material-based microfiber is used (optical fiber is used), from the textile manufacturing view point and with regard to mass production, what if plastic microfiber is used instead of optical fiber? As it is known that polyamide, polyester, viscose etc. are well known common fibers used in textiles and clothing because of their mechanical properties such as easy to process in manufacturing and, comfortable and durable enough for wearer because of flexibility, tensile, extension properties etc. In case the plastic microfiber should be chosen as substrate instead of optical fiber as suggested in page 11, then the novelty and the contribution of the paper would be clear and easily explained. Can it be concluded that your method might be also valid for plastic microfibers or not? Any remark on this? Or how do you explain your contribution to literature then?

-Moreover, the reliability and working range of miniaturized functional electronic devices onto optical fiber are not much discussed in the paper. The UV/thermal sensing behavior is not evaluated in detail. What are the reliability, sensitivity of the sensors, repeatability, hysteresis etc.? These issues should also be discussed.

-What about the abrasion during the production and usage? Fibers embedded to textile system will be definitely subjected to friction. Any results about this issue?

-Any extension and elongation results?

-The integration strategy of the fiber to PET substrate is not presented in detail. How did you do the sewing? Sewing is done manually or using a kind of sewing machine. I suppose the fiber can

not be used in sewing machine because of low mechanical properties. Please give explanation.

-The working performance of the fiber in terms of all mentioned criteria (transistor, thermoelectrical characteristics of temperature sensors, optoelectrical characteristics of UV sensors etc.) should also be reported after integration to collar of a shirt and after wearing for some period. Additionally, after the fiber is subjected to some environmental effects (e.g perspiration, abrasion) ? Please present results and give info.

-Moreover, if it is washable or not, or should be detachable etc. please add info.

-The square-shaped silica core microfiber ($150 \mu\text{m} \times 150 \mu\text{m} \times 7.5 \text{cm}$), is it drawn in your lab or commercially available, please give info?

-In the figures, generally schematic illustrations are given instead of real samples. It would be good to add real ones to convince the reader. Or any video demonstration would be also nice.

REVIEWER COMMENTS

Reviewer #1 (Remarks to the Author):

The English language of the paper is not vetted.

The notable result is to provide a fibre that integrates multiple components. this opens up prospects for future textile-electronic platforms. The proposed approach is interesting even if the final concept has properties that are a little different from the textile. The support remains rigid. This point will be improved for future generations. The approach is complementary to works like <https://doi.org/10.1038/s41467-021-23628-5>

[Response] Thank you for recognizing its interesting point. We are fully aware of the importance of the reviewer's comments regarding the readiness of our manuscript for publication. First of all, according to the reviewer's suggestion, we reevaluated the English language of our revised manuscript from Springer Nature Author Services (**The verification code 9DEA-083E-DA6F-8A79-6EDF**). To address the reviewer's concerns pertaining to our study and improve the quality of the present state in the manuscript, we sought to answer all the questions by including additional states and clarifying the discussion on the experimental results.

We emphasized the high-density integration and scalable multiplicity in our concept, which was more than enough to distinguish our work from previously published studies. The digital fibers by Y. Fink et al. were based on a fiber drawing process [R1]. Drawn fibers contain continuous domains connected to devices embedded along the entire length of the fiber. Hundreds of individually addressable digital devices are electrically connected in situ during the fiber drawing process but the gap distance between each device chip is ~6.5 μm . The polymer preforms with closely packed digital devices has a center-to-center distance of 0.65 mm between the devices. By contrast, analogue integrated circuits with fine patterns in our design can be fabricated on the surface of a microscale 3D single fiber by using conventional photolithography. Therefore, we believe that our electronic fiber platform is easier to integrate functional devices and reduce device area for high density thus to achieve a multiple functional electronic textile. According to the reviewer's comment, we have added reference [R1] as reference (17) and made a comparison with our work on page 4 in the revised manuscript.

- From this point of view, it is highly necessary to develop compact and miniaturized electronic systems that are capable of working on a single fibre. To impart multiple functions to the textile, the methods of inserting small electronic components into a fibre strand or yarn have been considered emerging candidates, enabling the implementation of a thermally drawn digital fibre and e-yarn.¹⁷⁻¹⁹ However, a limitation to the thermal drawing approach and the mounting of small components on the top surface of a filament is the low device density. A new strategy to fabricate a high-density electronic microfibre possessing multiple electronic components and circuits as well as maintaining excellent electrical performance has not yet been reported. (**on page 4**)

[R1] Loke, G. et al. Digital electronics in fibres enable fabric-based machine-learning inference. *Nat. Commun.* **12**, 3317 (2021).

The work presented appears rigorous and serious and is well presented. However, I have a few questions and remarks (minor revision):

1. Throughout the paper, avoid carriage returns between numbers and "fig.". (Line 111-112, 117-118, etc.)

[Response] We appreciate the reviewer's comment. We removed line breaks (carriage returns) in the revised manuscript to make it easier for the readers to follow.

2. The summary is well constructed and summarizes the main lines of the article in a clear and precise manner.

[Response] Thank you for evaluating our study in-depth. In order to clarify the summary (according to the reviewer #2' comment), we added Supplementary Fig. 10. and the following discussion on page 17 in the revised manuscript.

- To implement integrated electronic fibre by reel-to-reel process, precise and continuous control of the orientation of the fibre face is necessary to enable a continuous fabrication process during feeding the fibre. To overcome size and density limitation of integrated electronic components on a fibre, it is required to introduce higher resolution of maskless photolithography and modify their location of exposure modules enabling to use the entire circumference of fibre. Finally, the arrangement of the equipment for each process (e.g. coating, lithography, development, deposition, etching, and inspection) is also considered as an important factor to perform mass production of electronic fibre by continuous reel-to-reel process. (Supplementary Fig. 10).

3. Line 86: I don't understand the part of the sentence "...the cuboid shape silica fibre includes six planar faces in 3-axis...". The microfiber is square-shaped, therefore it has 4 planar faces. The ends are not exploited for the components (?).

[Response] We agree with this comment. We have made the change as suggested. We have revised the paragraph on page 5 in the revised manuscript for clarity.

- The microfibre substrate also features a 3D geometric shape including four planar faces throughout the length of the fibre that enables higher integration density. (on page 5)

4. Line 157: why a hyphen between "470" and "nm" (and "404" and "nm")?

[Response] We appreciate the reviewer's comment. We remove the hyphens between the numbers and the unit. We also revised the following paragraph in the revised manuscript. (on page 8, 19, 23 and 24)

- UV-LED light (wavelength: 470 nm) and UV-laser light (wavelength: 404 nm) irradiated both the top and bottom of the FET on a monofilament, implying UV sensing "out of fibre" and "through fibre core", respectively (Fig. 3a and 3d). (on page 8)
- Caption of Figure 3. (on page 19)
- METHODS section (on pages 23 and 24)

•

5. Line 177, 178: replace "A/W" with "A W⁻¹"

[Response] Thank you for helping us catch this mistake. According to the reviewer's comment, we replace "A/W" with A W⁻¹ on page 10 in the revised manuscript.

- The phototransistors presented a photoresponsivity of 0.64 A W⁻¹ at UV-LED light irradiated out of the fibre and 53.9 A W⁻¹ under UV laser light propagated through the fibre core, as shown in Fig. 3b and 3e. (on page 10)

6. Line 227: 0,64 and 0,68 % of strain is very low for textile fibre (as you write line 232). Why haven't you tested more? Do you know the limit, or the mechanical behaviour, of your microfiber substrate? It would have been interesting to carry out the mechanical tests to the limits and to make the electrical measurements at the same time.

[Response] We appreciate the reviewer's comment. The glass fiber substrate is relatively flexible compared to glass-based planar substrates owing to its thinness and small width. However, the glass microfiber is prone to fracture under high mechanical stress. In fact, the glass has a tensile modulus of ~69 GPa, which is an order of magnitude larger than those of plastic materials [Antunes, P. et al., Mechanical Properties of Optical Fibers, *Selected Topics on Optical Fiber Technology*, Rijeka, Croatia:InTech, (2012)]. The inherent mechanical property of the glass fiber leads to limited flexibility with restricted motion. The maximum bendability of our electronic fiber was measured through addition bending tests to the limits, as shown in Figure R1. We found that the threshold bending radius (R) of our electronic fiber was measured to be 7.3 mm, implying the calculated bending strain of 1.03%. As the bending radius (R) reached right before 7.3 mm, it still kept its flexibility. It means that the electronic fiber return to its pristine state as shown in Figure R1A. As shown in original Figure 5A, B, and C, the electronic components on the glass fiber kept their electrical characteristics up to a strain of 0.68 % (Bending radius (R) of fiber: 11.0 mm). We carried out electronic characteristics of the IGZO TFT under different bending conditions ($R=\infty$, 13 mm and 7.3 mm) at the same time. The calculated strain (ϵ) values were 0.58 and 1.03 % at $R = 13$ and 7.3 mm, respectively. As shown in Figure R1E-G, there were negligible changes in the electrical performance including field-effect mobility, on/off ratio, and threshold voltage. However, the electronic fiber (silica core) was broken when the bending radius (R) was less than 7.3 mm ($R < 7.3$ mm). Nevertheless, we would like to highlight that our 1D device has reasonable mechanical stability of the device performance under repeated bending cycles, even after 10,000 bending cycles at a fixed bending radius ($R = 11.7$ mm) (Fig. 5d in the original manuscript). Considering above results, the mechanical property of our electronic fiber platform are strongly related to that of fiber substrate. Hence, we think that the mechanical property may be still further improved in a future study by replacing both the substrate and the electrode with softer or more flexible materials.

For better understanding of the mechanical properties of our electronic fiber, we added Figure R1 (see next page) as Fig. S8. in the revised supplementary information and added the following sentences on page 13 in the revised manuscript.

- From the additional mechanical tests and its electrical characteristics of the electronic fibre to the limit, we found that the threshold bending radius (R) and strain (ϵ) of the electronic fibre were 7.3 mm and 1.03%, respectively, without the mechanical failure of the silica fibre. (Supplementary Fig. 8.) Because the electronic performance of the IGZO TFT on the fibre were maintained right before the breakdown of the fibre, the mechanical durability of our electronic fibre seems to depend on the mechanical properties of the fibre substrate. (on page 13)

Figure R1. Maximum bending test of the electronic fibre and electrical characteristics of the IGZO TFT on the electronic fibre. The bending radius (R) of (a) $R = \infty$, (b) $R = 13$ mm, (c) $R = 7.3$ mm, (d) $R < 7.3$ mm (Broken). The electronic fibre was broken at a bending radius of 7.3 mm. The calculated strain (ϵ) was 1.03 %. The electrical characteristics of the IGZO TFT on the electronic fibre at (e) $R = \infty$, (f) $R = 13$ mm, (g) $R = 7.3$ mm.

7. Line 248: For the perspectives, in addition to the change of support to work with a more flexible microfiber substrate, is it possible to work with a circular section fibre and to better exploit the entire circumference to increase the density of electronics components?

[Response] Thanks to the reviewer's kind comment. We also completely agree with the reviewer's comment. It is possible to work with a circular section fibre and use the entire circumference to increase the density of electronic components on the fibre surface. Especially, the density of electronic components is depending on the diameter of the fibre. If one used the larger fibre (larger circumference), more devices can be integrated. And also, it can be increased by scaling-down of the electronic component which enables fabrication of more devices on the limited area. As shown in Figure R2, we patterned different sizes of metal electrodes (5×5 , 10×10 , $20 \times 20 \mu\text{m}^2$) on a cylindrical glass fibre ($\phi = 125 \mu\text{m}$). It was possible to pattern more square electrodes on the surface of the cylindrical glass fibre. Unfortunately, we were not able to electronic components on the circular sectional fibre. As the reviewer knows, repeated alignment of pattern and thin film deposition is essential. Especially, positioning of the fibre substrate is critical to make alignment correctly. However, it is very difficult to identify bottom and top side of the circular sectional fibre in laboratory-scale experiments. On the other hand, a square sectional fibre was much easier to identify bottom and top side. Because it has four flat face, it was well placed and fixed on a sample holder without any rotation itself. Therefore, if someone do four times of integration process on four

different faces of the square sectional fibre, it is possible to use the entire circumference of the fibre. We believe that our approach is exactly the same perspective as the reviewer's comment. Based on the reviewer's comment, it is worth mentioning the density of the electronic fiber in the revised manuscript. We added the following sentences on page 5 in the revised manuscript.

- The microfibre substrate also features a 3D geometric shape including four planar faces throughout the length of the fibre that enables a higher integration density. The most efficient way to increase the density of electronic components is to use the entire circumference of fibre. In order to investigate the feasibility of this, we integrated electronic components on two different sides of the square sectional fibre. (on page 5)

Figure R2. Optical microscope and SEM images of various metal patterns on a microfibre. Top: The optical microscope images of square electrode pattern arrays having various sizes (5×5 , 10×10 , $20 \times 20 \mu\text{m}^2$) and pitches fabricated on cylindrical glass fibre ($\varphi = 125 \mu\text{m}$). Bottom: The optical microscope and SEM images of various patterned metal electrodes on cylindrical glass fibre.

Reviewer #2 (Remarks to the Author):

Review: Integration of multiple electronic components on a microfiber toward emerging electronic textile platform

Overall, this is a very interesting paper with a potential platform fabrication process which would benefit e-textile development. I would recommend publication but with some changes as listed below.

This paper would benefit from a native English speaker to improve the grammar; some of the sentences are clunky and make reading, what is otherwise an interesting paper, difficult at times. Particular focus should be given to the appropriate use of 'the' which is often missing. Despite this minor comment, it is well written overall and the principle discussion is easy to follow.

[Response] Thank you for the reviewer's positive comment. We revised several wordings in the manuscript accordingly. This paper has been revised by professional English editing service. (Springer Nature Author Services, <https://authorservices.springernature.com>). (**The verification code 9DEA-083E-DA6F-8A79-6EDF**).

I believe the noteworthy result is the ability to fabricate basic circuits on the very fine fiber/yarn which could enable complex circuitry to be embedded into e-textiles. The literature review is good but would benefit with the inclusion of some more contemporary e-textile yarn comparisons, for example the work at Imperial College London, Nottingham Trent Advanced Textile Research Group, University of Southampton and University of Glasgow. Their work is competitive with existing electronic components embedded into e-yarns or filaments; I believe this paper is taking this concept further with circuits fabricated on the yarn itself which is impressive.

[Response] We thank the reviewer for the critical reading of the manuscript and for prompting us to expand on some important aspects of the work. We have emphasized the ability to fabricate basic circuits on the very fine fiber, the high-density integration and scalable multiplicity in our concept, which is distinction of our work from previously published studies. Analogue integrated circuits with fine patterns in our fiber platform can be fabricated on the surface of a microscale 3D single fiber by using conventional photolithography. Therefore, we believe that our electronic fiber platform is easier to integrate functional devices and reduce device area for high density thus to achieve a multiple functional electronic monofilament. According to the reviewer's comment, we have added the reference (17–19) and added the following sentences to make a comparison with our work on page 4 in the revised manuscript

- From this point of view, it is highly necessary to develop compact and miniaturized electronic systems that are capable of working on a single fibre. To impart multiple functions to the textile, the methods of inserting small electronic components into a fibre strand or yarn have been considered emerging candidates, enabling the implementation of a thermally drawn digital fibre and e-yarn. However, a limitation to the thermal drawing approach and the mounting of small components on the top surface of a filament is the low device density. A new strategy to fabricate a high-density electronic microfibre possessing multiple electronic components and circuits as well as maintaining excellent electrical performance has not yet been reported. (**on page 4**)

[R1] Loke, G. et al. Digital electronics in fibres enable fabric-based machine-learning inference. *Nat. Commun.* **12**, 3317 (2021).

[R2] Lugoda, P. et al. Flexible temperature sensor integration into e-textiles using different industrial yarn fabrication processes. *Sensors* **20**, 73 (2020).

[R3] Komolafe, A. et al. Integrating flexible filament circuits for e-textile applications. *Adv. Mater. Technologies* **4**, 1900176 (2019).

I have some questions on the fabrication methodology which appears sound but it would be good to have more detail on how this could be scaled up and the current size limitations but these are contained within my points below.

I have some specific comments:

Line 54 – please explain what you mean by “any patterning on as-processed” do you mean there is no patterning process during manufacture?

[Response] Thanks for pointing this out. We have revised the sentence to make it easier for readers to understand on page 3 in the revised manuscript.

- The existing electronic fibre platforms are generally composed of only one type of electronic component with a single function on a fibre substrate that is attributed to all around wrapping of an active layer on the entire fibre without patterning at the desired area on the surface of the fibre during the manufacturing process. **(on page 3)**

Line 87 – what temperature range can it sustain? “high temperature” should be defined – e.g. 90 °C is a high temperature for fabrics, but a low temperature for most electronic fabrication methods.

[Response] Thank you for your kind comment. As we described in the original manuscript, the fiber is composed of fused silica. It is well known that the fused silica can sustain up to 1100 °C without melting. Therefore, both inorganic and organic electronic materials can be deposited on the fiber substrate. The baking temperature of the substrate during photolithography and ALD is a maximum 100 °C, and IGZO films were thermally annealed on a hot plate at 300 °C in ambient air. Although the processing temperature is high for fabric, the thread-like electrical fiber can be fabricated using a reel-to-reel process and then embedded into fabrics at room temperature. The process temperature may be further reduced by introducing a low-temperature process (<150 °C) such as deep ultraviolet (DUV) photochemical activation and sol-gel process of metal-oxide dielectrics and semiconductors [Jo, J.W. et al., *Adv. Mater.* **27**, 1182-1188 (2015)]. Park et al. demonstrated that the DUV-annealed metal-oxide TFTs and circuits with optimized sol-gel ZAO dielectric and IGZO semiconductor layers are high performance (saturation field-effect mobility: 20 cm²V⁻¹s⁻¹) and electrically/mechanically stable even on an ultra-flexible medium. It exhibits that scalable high-performance flexible electronics fabricated via solution-processed inorganic materials at low temperature is readily possible [Jo, J.W. et al., *Adv. Mater.* **27**, 1182-1188 (2015)]. Nevertheless, we need to further demonstrate through experimental validation that the proposed low-temperature solution process can be used in this fiber platform with high device yield, high device performance, low leakage current, and environmental stability.

According to the reviewer’s comment, we revised the following sentence on page 5 in the revised manuscript.

- Although the silica-based fibre has relatively lower flexibility than polymeric substrates, it can sustain high process temperature of 1100 °C without melting, allowing high-performance inorganic electrical materials to be deposited on it. **(on page 5)**

Line139 – do you have any information on the yield of these circuits; I realise these are prototypes so any yield should be taken in that context but it would be good to know how difficult this process is to achieve.

[Response] As the reviewer’s mentioned, it is difficult to achieve high-yield production at the early stage of fabrication due to many process steps and contamination, which can change device properties.

At the initial state, we got acceptable transfer curves less than 3 from total 8 transistors on 10 cm long electronic fiber. Therefore, the yield was less than 40%. Finally, we achieved almost 70% device yield by counting working transistors. A comprehensive yield can be evaluated by measuring the electrical properties of transistors fabricated on the fiber substrate. The IGZO FETs on the monofilament have yielded average field-effect mobility of $5.5 \text{ cm}^2\text{V}^{-1}\text{s}^{-1}$ and the standard deviation of $1.1 \text{ cm}^2\text{V}^{-1}\text{s}^{-1}$, as probed on five FETs for each electrical monofiber. The threshold voltage and the sub-threshold swing are $0.28\pm 0.57 \text{ V}$ and $0.36\pm 0.11 \text{ V dec}^{-1}$. The ideal electrode pattern without cracks allows the FETs to operate with low leakage and to be fabricated with good uniformity. However, due to difficulty in fabrication of electronic circuit on a fiber, we achieved only 60 and 40% of yield for inverter and RO, respectively.

According to the reviewer's comment, we revised the following paragraph on page 7 in the revised manuscript.

- At the initial stage of fabrication, we got acceptable transfer curves less than 3 from the total of 8 transistors on 10 cm long proto-type electronic fibre, indicative of less than 40% of yield. Finally, we achieved almost 70% device yield by counting working transistors. Subsequently, five individual FETs were fabricated on each monofilament that exhibited average saturation mobility of $5.5 \text{ cm}^2\text{V}^{-1}\text{s}^{-1}$ with a standard deviation of $1.1 \text{ cm}^2\text{V}^{-1}\text{s}^{-1}$, threshold voltage (V_{Th}) of $0.28\pm 0.57 \text{ V}$, and low subthreshold swing of $0.36\pm 0.11 \text{ V dec}^{-1}$. **(on page 7)**
- Due to the complicated fabrication procedure and design of electronic circuits on fibre, we achieved only 60 and 40% of yield for inverter and RO, respectively. **(on page 7)**

Line 143 – Is there any difference in device quality when fabricating on the rectangular or cylindrical fibre?

[Response] A cylindrical fiber has a curved surface, of which the center and the edge are exposed to a laser light beam for direct lithographic patterning of the photoresist. However, the intensity of exposure varies at different points along the curvature that leads to inconsistent in pattern linewidths and broadening of patterns. The present technique can produce the undesired pattern by using a standard aligner, which controls motion along the x- and y-axis but cannot rotate around the z-axis. Therefore, the patterning technique for the cylindrical fiber substrate should be further modified for an accurate and fine electrode pattern.

Line 151 – Did you perform any measurements of the UV sensor response once it is embedded in the textile? Please comment on the potential issues, if any, that you envisage – for example you would be unable to control the orientation of the fiber in the garment.

Line 158 – Linked with the above, for the out of fiber and through core measurement, you would not be able to control the direction once embedded in a garment, so can you separate these two measurements out at all to therefore determine the intensity of UV regardless of direction?

[Response] We appreciate the reviewer's comment. Here, we answer above two questions. When the electronic fiber embedded in a garment (textile), it was very difficult to evaluate UV sensor response both the out of fiber and through core measurement as shown in Figure 3a and d in the original manuscript. For through core measurement, it is necessary to make a precise junction between electronic fiber core and a core of the optical waveguide (optical fiber) that is connected to the laser diode. Due to limitations of the measurement system setup, we were not able to make good junction between them to allow optical signal (UV laser) thorough the core properly. We frequently observed breakdown of the electronic fiber during sawing right after formation of junction between electronic fiber and optical fiber. To prevent unexpected breakdown and control the orientation of the fiber in the garment during sewing, we introduced a special sewing method (syringe needle assisted sewing

method, See Video R1(Supplementary Video 1)). However, we were not achieved a proper measurement setup and unable to measure the optical signal through core. On the other hand, we successfully sewed individual electronic fiber in a garment using the syringe needle assisted sewing method. Because the syringe needle guides the electronic fiber during sewing, we were able to control the orientation of the fiber in the garment, resulting in the UV sensor upward for both electrical contacts for UV sensing and uniform UV illumination.

According to the reviewer’s comment, we carried out optoelectrical characteristics of the electronic fiber after integration to the collar of a shirt and after wearing for 10 days, as shown in Fig. R3 and Supplementary Fig. 9. The photocurrents of the electronic fibers were extracted from the transfer curves at $V_G = -5$ V and $V_D = 0$ V, under 470 nm LED light irradiation out of the fiber, which was the same measurement condition of Figure 3A in the revised manuscript. According to the reviewer’s comments, we added the following paragraph on pages 14-15 and put Figure R3 as Fig. 5g in the revised manuscript.

- To prevent unexpected damage to the electronic fibre during the sewing process, we introduced a special sewing method by using a needle of a syringe that enables the fibres to place in the fabric safely (Supplementary Mov. 1). It helped us to sew the electronic fibre in the fabric without any serious damage to the surface. Because the syringe needle guides the electronic fibre during sewing, we were able to control the orientation of the fibre in the garment(bandage), resulting in the UV sensor or temperature sensor upward for both electrical contacts for UV or temperature sensing. **(on page 14)**
- We monitored the switching performance of the IGZO FET, optoelectrical characteristics of the UV sensor, and thermoelectrical characteristics of the temperature sensor on the fibre at pristine, sewed and after wearing for 10 days (Supplementary Fig. 9). The IGZO FET maintained its electrical performance for 1 day. After 10 days, it exhibited a slightly degraded field-effect mobility above $7.4 \text{ cm}^2\text{V}^{-1}\text{s}^{-1}$ (Fig. 5f). Additionally, we observed a slight decrease in drain current from 2.9 to 1.8 μA and a positive shift in threshold voltage from -2.4 to -1.7 V, respectively. From the additional measurements of the photocurrent and thermoelectric voltage of the electronic fibre before and after wearing for 10 days, as shown in Fig. 5g, we found that the photosensor was relatively stable, maintaining photocurrent values of 15.3 and 13.7 μA for the LED on and off states, respectively, at $V_G = 0$ V and $V_D = 5$ V. **(on page 14-15)**

- **Figure R3.** Photocurrent of photosensor and thermoelectric voltage of TCs on the fiber at

pristine, as sewed, and after wearing for 10 days.

Concerning to the question on “*for the out of fiber and through core measurement, you would not be able to control the direction once embedded in a garment, so can you separate these two measurements out at all to therefore determine the intensity of UV regardless of direction?*”, we also agree to the reviewer’s comment. As answered above, we were barely able to control the orientation of the electronic fiber embedded in a garment this time by using a special sewing method. However, this is the only way to demonstrate the feasibility of our electronic fiber as one of an emerging integrated electronic fiber platform. Generally, if the electronic fiber is completely buried in a garment, it is difficult to perfectly detect the intensity of UV light illuminated from all directions. The textile on the device limits the active area, which decreases optical absorption and lowers space utilization. To overcome these challenges, the transparent glass fiber substrate offers structural advantages that surmount limitations faced by wearable optoelectronic devices on planar substrates. To improve the omnidirectional detection capabilities of optoelectronic devices, researchers usually employ antireflection layers, such as NW arrays and resonant plasmonic and metamaterial structures [Lien, D. H. et al., *npj Flexible Electron.* **2**, 1-7 (2018)]. However, an inherent limitation is that non-transparent planar devices can only detect incident light from one-half of ambient space (i.e., the front 180° field of view). To detect light behind the device (in the rear 180° field of view), it is necessary to use sophisticated lenses, prisms, and other optical components, which makes the system cumbersome [Lien, D. H. et al., *npj Flexible Electron.* **2**, 1-7 (2018)]. By using a 1D substrate as a transparent, flexible, and optical fiber, our device enables 360° omnidirectional photodetection. When a series of optical pulses are injected from the end of fiber and light propagates inside the optical fiber, the UV sensor detects scattered light (Figs. 3D-F). (Note that the intensity of the scattered light attenuates with distance, it is necessary to place the sensor as close to the leakage point in the optical fiber as possible.) Moreover, incident UV light from an oblique angle can be detected through the three-dimensionally transparent fiber substrate (Figs. 3A-C). Although our device doesn’t have an extremely accurate way of detecting its surroundings in the textile, the ability to detect light from an oblique angle is needed for optoelectronic applications in deformable fabrics. According to the reviewer’s comment, we added following discussion on page 15 in the revised manuscript.

- Although our electronic fibre in a garment successfully detected UV signal out of fibre, it is due to the syringe needle-assisted sewing method. If the electronic fibre is completely buried in a garment by the conventional sewing method, it is hard to expect proper detection of UV light illuminated from all directions in general. Therefore, further research on how to place electronic fibre in a garment and evaluate its sensing performances is still required for practical UV sensing electronic fibre applications. **(on page 15)**

Line 161 – this is unclear “was responded against”

[Response] Thanks for pointing this out. We have revised the sentence to make it easier for readers to understand on page 9 in the revised manuscript.

- Upon illumination, there is a significant increase in off-current from 4.0×10^{-8} A to 7.5×10^{-7} A at $V_G = 0$ V. This implies that exposed light contributes generation of photocarriers in the IGZO channel, inducing higher channel conductivity. **(on page 9)**

Line 183 – Some excellent examples of potential applications and a platform technology for more complex circuitr/sensors in the future. However, it is not clear how big the circuits could be in terms of continuous length along the fiber, e.g. not just a string of individual circuits but a single interconnected circuit?

[Response] We appreciate the reviewer’s comment. an electronic fiber (length: 10 cm) containing

approximately 30 interspersed ICs, inverters, photosensors, transistors, condensers, and temperature sensors have been demonstrated as a proof of concept. The entire devices are individually and independently operated but the integration of a photosensor, an inverter, and a transistor can fulfill more than one function simultaneously. Due to the interior size of lab-scale manufacturing equipment, the length of fiber substrates is limited. Nevertheless, a roll-type photolithography [Lee, S. H. et al., *J. Micromech. Microeng.* **26**, 115008 (2016)] and a continuous process system [Loke, G. et al. *Nat. Commun.* **12**, 3317 (2021); Lugoda, P. et al. *Sensors* **20**, 73 (2020)] may be capable of scalable manufacturing of these electrical fibers for fabrication of continuously interconnected circuits on a monofilament. In this sense, in terms of the potential of the fiber platform technology, we believe that this conceptual demonstration of electrical fiber is still meaningful.

According to the reviewer's comment and for a better understanding of our electric fiber, we put the following sentences on page 6 in our revised manuscript.

- The integrated electronic fibre (length: 10 cm) containing approximately 30 interspersed ROs, inverters, phototransistors, condensers, and temperature sensors have been demonstrated as a proof of concept. All devices are able to operate individually and independently. **(on page 6)**

Line 207 – How would the readings be affected by being inside a fabric? In this scenario the heat block would not be at the end but would be in parallel to the thermocouples, do you envisage any problem or would they all just give the same reading in such a close area?

[Response] We appreciate the reviewer's comment. As the reviewer mentioned, thermocouples (TCs) on the fiber will be placed in the same position inside a fabric. Therefore, the almost same reading value of thermoelectric voltages is expected at different positions of TCs. To verify this, we performed thermoelectrical characteristics of each TC on the fiber, which was placed on a hot chuck. At room temperature, all TCs exhibited an almost same value of around 0.5 μV of thermoelectric voltage. By increasing the set temperature of hot chuck up to 40 $^{\circ}\text{C}$, there were negligible differences in thermoelectric voltages at each TC (TC1, TC2, TC3), exhibiting the almost same value of 135 μV . For comparison, we did the additional test to evaluate the performance of TCs inside a fabric placed on a hot chuck. All TCs exhibited almost the same value of approximately 0.52 μV of thermoelectric voltage at room temperature (23 $^{\circ}\text{C}$). And, the average thermoelectric voltage of each TC (TC1, TC2, TC3) in the fabric showed a similar value of approximately 128.3 μV at 40 $^{\circ}\text{C}$. From the above results, we concluded that each TC at the same position from the heat source (hot plate) showed negligible differences in their performances. In addition, the average thermoelectric voltage of each TC (TC1, TC2, TC3) showed a relatively higher value than that of TCs being in the fabric due to better transfer of heat from hot chuck to TCs on the fibre. According to the reviewer's comment, we added the following sentences on page 15 and put Figure R4 as Fig. 5g in the revised manuscript.

- Additionally, we evaluated the thermoelectrical characteristics of each TC on the fibre in the fabric placed on a hot chuck. All TCs exhibited almost the same value of approximately 0.52 μV of thermoelectric voltage at room temperature (23 $^{\circ}\text{C}$) before and after wearing for 10 days. Similarly, the average thermoelectric voltage of each TC (TC1, TC2, TC3) showed a similar value of approximately 128.3 μV at 40 $^{\circ}\text{C}$ before and after wearing for 10 days. **(on page 15)**

- **Figure R4.** Photocurrent of photosensor and thermoelectric voltage of TCs on the fiber at pristine, as sewed, and after wearing for 10 days.

Line 208 – would it be better for the e-textile to just have one large sensor rather than 3 individuals?

[Response] We thank the reviewer for this comment. As the reviewer knows, e-textile will be used by being inside fabric for practical application. In this point of view, we also strongly agree that one large temperature sensor will be a good approach. However, the main reason why our temperature sensor was designed on a monofilament is to make sure the validity of our e-textile platform. As shown in Figure 4, the distance between Cr-Ni thermocouple (TC) junctions is 3.4 mm. If TCs are working well, we thought that they exhibit different thermoelectric voltages (ΔV_{TE}) depending on the distance from the heat source (hot block). Based on this hypothesis, we integrated TCs (TC1, TC2, TC3) on a fiber in the longitudinal direction. As mentioned above, the TCs showed almost the same thermoelectric voltages of 0.5 and 135 μV at room temperature and 40 °C of hot chuck, respectively. As part of future work, discrete sensors can also be integrated into the fiber, which allows for the collection of distance-dependent thermal information.

Line 229 – unclear text – “was kept its device parameters”

[Response] Thanks for pointing this out. We have revised the sentence to make it easier for readers to understand on page 13 in the revised manuscript.

- The IGZO FET on the fibre sustained 10,000 cycles of repeated bending at the bending radius of 11.7 mm. We did not observe apparent breakdown of the fibre or delamination of the semiconductor or metal electrode during or after the bending test. The saturated mobility (μ_{sat}) and threshold voltage (V_{Th}) of the FET slightly decreased from 3.77 to 3.73 $cm^2V^{-1}s^{-1}$ and $-0.75 V$ to $-0.81V$, respectively. The drain current ($I_{D,on}$) of the FET was measured to be $\sim 1.28 \mu A$ at each bending condition with negligible changes, as shown in Supplementary Fig. 8. **(on page 13)**

Line 235 – you say conventional semiconductor fabrication process – please give more detail on any modifications required or size limits when using this approach with your particular system and future potential scale up in both production volume and device/system size.

[Response] We appreciate the reviewer’s comment. The size of ICs and devices was limited due to the demonstration in lab-scale experimental production systems. The translation of the lab-scale method to continuous manufacturing processes is indispensable for the adaptation of this fiber platform to applications, including large-scale production and high density of the device. For example, a roll-type photolithography [Lee, S. H. et al., *J. Micromech. Microeng.* **26**, 115008 (2016)] and a continuous reel-to-reel process [Loke, G. et al. *Nat. Commun.* **12**, 3317 (2021); Lugoda, P. et al. *Sensors* **20**, 73 (2020)] may be capable of scalable manufacturing of these electrical fibers. Although the mass production of chip on a fiber is presently hampered by relatively slow manufacturing speeds of laser patterning, further advancement of the reel-to-reel process (see Fig. R5.) could demonstrate scale-up in production volume and device/system size.

According to the reviewer’s comment, we added Fig. R5 as Supplementary Fig. 10 and the following discussion on page 17 in the revised manuscript.

- To implement integrated electronic fibre by reel-to-reel process, precise and continuous control of the orientation of the fibre face is necessary to enable a continuous fabrication process during feeding the fibre. To overcome size and density limitation of integrated electronic components on a fibre, it is required to introduce higher resolution of maskless photolithography and modify their location of exposure modules enabling to use the entire circumference of fibre. Finally, the arrangement of the equipment for each process (e.g. coating, lithography, development, deposition, etching, and inspection) is also considered as an important factor to perform mass production of electronic fibre by continuous reel-to-reel process. (Supplementary Fig. 10). (on page 17)

Figure R5. Schematic of reel-to-reel process for mass production of electronic fibre. The photographs of electrode patterns formed on a monofilament during CTAC process, maskless photolithography, development, metal deposition, and wet etching process.

Line 246 – please elaborate on ‘plastic microfiber’ alternatives; you previously said you were using fused silicon because it allowed the use of inorganic material deposition, so which material are you suggesting and would this affect this requirement?

[Response] Thanks for pointing this out. There are several technologies available for the fabrication of electronics on flexible substrates. M. Ahmed et al. demonstrated an ultrathin, flexible surface-mount sensor circuit [Ahmed, M. et al. *IEEE Sens. J.* 12, 864 (2012); Ahmed, M. and Butler, D. P. *J. Vac. Sci. Technol. B, Microelectron. Nanometer Struct.* 31, 050602 (2013)]. H. Lin et al. investigated the thermal and electrical conduction in the Au films deposited on the polyimide fiber, showing great potential in the flexible microelectronic device field as a flexible substrate [R4]. The glass transition temperature of the polyimide substrates employed, ~ 400 °C, does limit the processing temperature that can be used [Ahmed, M. and Butler, D. P. *J. Vac. Sci. Technol. B, Microelectron. Nanometer Struct.* 31, 050602 (2013)]. Polyimide, one of the flexible polymeric materials, has been used as flexible fiber substrates for the roll-type production of electronic fibers since it is an organic material with high performance, which possesses a series of great features, such as low thermal conductivity, high tensile strength, tensile modulus, thermal stability, chemical stability, radiation resistance, and insulativity [R4]. Therefore, the flexible polymeric filament can be considered as one of the substitutional fiber substrates for better flexibility and processability. As another example, we prepared a square-shaped filament of SU8 that has a dimension of $100\ \mu\text{m} \times 100\ \mu\text{m}$. Due to the limited processing temperature of the SU8 monofilament, we could not fabricate an IGZO-based field-effect transistor but prepared electrode patterns on the SU8 filament. From the I - V and resistance characteristics of the metal electrode on the fiber substrate, we identify that there were negligible differences in electrical property between before (flat) and after bending (coiling it around a glass cylinder with a diameter of 1.3 mm), as shown in Figure R6. We conclude that the electrode on a plastic fiber well maintained its electrical property.

Figure R6. Photograph of the metal electrodes on the plastic fiber (a) flat and (b) coiled on a capillary tube with a diameter of 1.3 mm. (c) I - V characteristics and resistance of the metal electrodes on the plastic fiber on both flat and coiled.

We have revised the sentence to make it easier for readers to understand on page 13-14 in the revised manuscript.

- Due to the limitation of the mechanical properties of inorganic material-based microfibres, it was difficult to achieve better flexibility than that of conventional flexible electronic devices. However, it is essential to use high-performance inorganic semiconducting materials to implement high-performance and integrated electronic fibre system. Especially, silica-based microscale fibre can be considered as one of the promising fibre substrates due to its

capability of achieving better electronic properties and scaling down by introducing both high-temperature and conventional semiconductor fabrication processes. Nevertheless, it is still required to develop better fibre substrate that enables both high performance and flexibility integrated electronic fibre system. For example, flexible polymeric materials, such as polyimide, include a combination of excellent properties, such as chemical stability, thermal stability, low thermal conductivity, radiation resistance, insulativity, high tensile strength and tensile modulus, which can be considered as substitutional fibre substrates for better flexibility and processability. [R4] (on page 13-14)

[R4] Lin, H. et al. Characterization of thermal and Electrical Transport in 6.4 nm Au films on polyimide film and fiber Substrates. *Sci. Rep.* **10**, 1–9 (2020)

Additional general comments:

1. It's not entirely clear which is the main novelty being claimed in the paper – is it the ability to process these devices on the fiber, is it the process itself or the idea of using it for e-textiles? It does feel as if the e-textile part has just been tacked on because the integration of this is only mentioned in Fig1A and doesn't appear to be tested beyond the concept? E-textile is in the title so more should be added to discuss the requirements for this and any tests or knowledge that has been obtained to identify these fibers as suitable.

[Response] Thank you for your valuable comments. New designs and technologies that can integrate compact devices onto a monofilament, which is an alternative thread (or a single yarn), are needed. In this sense, we have developed a new fiber platform and process for the reduction of device interconnections that also allows for discrete devices with various configurations on a single fiber. Electrode patterning through maskless photolithography enables the interconnection and integration of electronic components on a single fiber while each electronic fiber can be connected by wiring or weaving conductive fibers.

To demonstrate feasibility for a potential e-textile application, the electronic fiber was directly sewed in a piece of common compression bandage. And then, the fabric was sewn inside of the collar of a shirt again, without any additional protective coating as shown in Fig. 5e and Supplementary Fig. 9a. We monitored the switching performance of the IGZO FET on the fiber at pristine, as sewed and after wearing for 10 days (Supplementary Fig. 9). The device operated well without serious malfunction after 10 days. Although our electronic fiber worked well above stressful conditions, we should note here that additional protecting or shielding layer is required to eliminate expected risks related to their contact with the human body while keeping the electrical function of the devices against various environmental conditions (mechanical stress, chemicals, and sweat etc.). Therefore, we believe that our electronic fiber platform is still considered as one of valid approaches for an integrated electronic textile system. However, due to the limited photolithography and weaving technologies in our manufacturing system, the wearable applications and the size of electronic fibers that we can fabricate were limited.

Additionally, we agree with the reviewer's concern that this is insufficient for discussing the requirements of e-textile. According to the reviewer's comment, we added the following sentences on additional discussion wearability of our electronic fiber on page 14-16 in the revised manuscript.

- Wearable e-textiles should be breathable and washable with high flexibility and shape adaptability. To demonstrate feasibility for a potential e-textile application, the electronic fibre was directly sewed in a piece of common compression bandage. To prevent unexpected damage to the electronic fibre during the sewing process, we introduced a special sewing method by using a needle of a syringe that enables the fibres to place in the fabric safely (Supplementary Mov. 1). It helped us to sew the electronic fibre in the fabric without any serious damage to the surface. Because the syringe needle guides the electronic fibre during sewing, we were able to control the orientation of the fibre in the garment(bandage), resulting in the UV sensor or temperature sensor upward for both electrical contacts for UV

or temperature sensing. Then, the fabric was sewn inside of the collar of a shirt again without any additional protective coating, as shown in Fig. 5e. We monitored the switching performance of the IGZO FET, optoelectrical characteristics of the UV sensor, and thermoelectrical characteristics of the temperature sensor on the fibre at pristine, sewed and after wearing for 10 days (Supplementary Fig. 9). The IGZO FET maintained its electrical performance for 1 day. After 10 days, it exhibited a slightly degraded field-effect mobility above $7.4 \text{ cm}^2\text{V}^{-1}\text{s}^{-1}$ (Fig. 5f). Additionally, we observed a slight decrease in drain current from 2.9 to 1.8 μA and a positive shift in threshold voltage from -2.4 to -1.7 V, respectively. From the additional measurements of the photocurrent and thermoelectric voltage of the electronic fibre before and after wearing for 10 days, as shown in Fig. 5g, we found that the photosensor was relatively stable, maintaining photocurrent values of 15.3 and 13.7 μA for the LED on and off states, respectively, at $V_G = 0 \text{ V}$ and $V_D = 5 \text{ V}$. Although our electronic fibre in a garment successfully detected UV signal out of fibre, it is due to the syringe needle-assisted sewing method. If the electronic fibre is completely buried in a garment by conventional sewing method, it is hard to expect proper detection of UV light illuminated from all directions in general. Therefore, further research on how to place electronic fibre in a garment and evaluate its sensing performances is still required for practical UV sensing electronic fibre applications. Additionally, we evaluated the thermoelectrical characteristics of each TC on the fibre in the fabric placed on a hot chuck. All TCs exhibited almost the same value of approximately 0.52 μV of thermoelectric voltage at room temperature (23 °C) before and after wearing for 10 days. Similarly, the average thermoelectric voltage of each TC (TC1, TC2, TC3) showed a similar value of approximately 128.3 μV at 40 °C before and after wearing for 10 days. Although our electronic fibre worked well above stressful conditions, we should note here that an additional thick protecting or shielding layer is required to eliminate expected risks related to their contact with the human body while keeping the electrical function of the devices against various environmental conditions (mechanical stress, chemicals, sweat, etc.).^{30,31} To further evaluate the washability of the electronic fibre, we carried out the CTAC process (speed 1.0 mm min⁻¹) of SU-8 solution to form a passivation layer (thickness = 2 μm). The electronic fibre was completely covered by an SU-8 passivation layer. Then, the encapsulated electronic fibre was dipped in detergent solution (5 ml in 90 ml of tap water) and NaCl solution (0.5 wt% for artificial human sweat) for 30 min and rinsed in pure tap water with stirring at 600 rpm at room temperature.³² After washing, the electronic fibre was dried at 60 °C on a hotplate. Because the SU-8 layer completely covered the outer electronic fibre, the electronic performance of the IGZO FET showed a negligible difference in transfer characteristics (field-effect mobility of $3.74 \text{ cm}^2\text{V}^{-1}\text{s}^{-1}$ in the saturation regime and an on/off current ratio of 4 orders of magnitude) before and after washing with detergent and 0.5 wt % NaCl solutions, as shown in Fig. 5h. This implies that the encapsulated electronic fibre maintained a stable performance, regardless of the wet environment, such as washing and perspiration conditions. It may be possible to implement practical electronic fibres by introducing a reliable protecting or encapsulation layer that is durable under various mechanical or chemical conditions. Therefore, we believe that our electronic fibre platform is still considered a valid approach for integrated electronic textile systems. (on page 14-16)

2. *It is more conventional in textile engineering to define the substrate as a yarn in your case, fiber would refer to all the smaller fibers that are then spun together to form the yarn. I would recommend either changing it throughout or might be easier to put a caveat at the beginning to explain your choice of terminology, e.g. "in this paper we define a 'fiber' as..." and relate that to conventional textile terminology.*

[Response] We appreciate the reviewer's comment. By definition, yarn is a long continuous length of interlocked fibers, suitable for use in the production of textiles, sewing, crocheting, knitting, weaving, or embroidery. Fiber is defined as a long and thin filament that can be embedded into fabric. Recently, many researchers have reported using the word 'fiber', meaning a thin filament [Xiong, J. et al. *Adv.*

Mater. **33**, 2002640 (2020); Shi, J. et al. *Adv. Mater.* **32**, 1901958 (2019); Chatterjee K. et al. *Adv. Mater.* **32**, 1902086 (2019)].

We have made the change as suggested. We have added the words such as a ‘filament’ and a ‘monofilament’ for clarity on page 4 in the revised manuscript.

- In this work, we present a new electronic fibre platform that enables LSI of electronic device components on the surface of a 1D fibre, defined as a monofilament with a diameter of 150 μm (Fig. 1a). (on page 4)

3. Fused silica is a relatively stiff ceramic compared to traditional textile or e-textile materials; how flexible is any fabric containing these microfiber yarns?

[Response] We appreciate the reviewer’s comment. When a mechanically homogeneous substrate of thickness d is bent to a cylindrical radius r , perpendicularly to the axis of bending, its outside surface expands and its inside surface is compressed by the bending strain $\varepsilon = d/2r$. A conventional approach to keeping ε low even in sharp bending, to small r , is to make the structure thin. In this way, the strain experienced by the active devices in bendable or rollable electronics can be kept small, particularly when the devices are placed in the neutral plane [Wong W. S., Salleo A., eds. *Flexible Electronics: Materials and Applications* (Springer, 2010)]. Although fused silica features stiff and brittle, it becomes flexible when made thin. Coating passivation (encapsulation) materials on the electronic fiber is a way to protect electronic fiber against various deformations. Because we have used a square silica core with a dimension of 150 $\mu\text{m} \times 150 \mu\text{m}$, it requires hundreds of micrometers of the outer buffer layer to protect the fiber effectively by locating the fiber at the mechanical neutral plane [Kim, D. H. et al., *Adv. Mater.* **22**, 2108–2124 (2010)]. To explain the effectiveness of the outer protective layer, a knot was made using bare optical fiber to indirectly check how effectively the thick outer layer protects the fiber. As shown in Figures R7 and R8, the bare optical fiber has thick outer protective layers with thicker than 150 μm , exhibiting that the fiber was not broken under a bending radius of 5 mm (the calculated strain $\varepsilon = 5.0\text{--}5.5\%$). From the above results, we conclude that both plastic fiber and the introduction of a protective layer should be considered for practical use of electronic fiber with better reliability. Although we did not take into consideration a thick passivation or shielding layer on the outer of the electronic fiber at the present stage due to difficulty in the probe, we are fully aware of the necessity of encapsulation of the fiber for practical wearable application. According to the reviewer’s comment, we added following sentences to discuss on how flexible is any fabric containing these microfibers on page 16 in the revised manuscript.

- This implies that the encapsulated electronic fibre maintained a stable performance, regardless of the wet environment, such as washing and perspiration conditions. It may be possible to implement practical electronic fibres by introducing a reliable protecting or encapsulation layer that is durable under various mechanical or chemical conditions. (on page 16)

Figure R7. The structure of the optical fiber. The optical fiber features a 150 $\mu\text{m} \times 150 \mu\text{m}$ of square

silica core, 250 μm of hard polymer cladding diameter, and 500 μm of Tefzel buffer coating diameter. Note that this image is from <https://www.thorlabs.com/>

Figure R8. The optical fiber and its overhand knot.

4. There is no detail on how these devices are connected to, either for testing or for use in a garment; has this been considered, please comment even if it is future work.

[Response] We appreciate the reviewer's kind comment. The electrical fiber is composed of discrete devices along with the same monofilament. Conventional tungsten probe tips are used for contact with the electric fiber, as shown in Figure R9. To verify a potential e-textile application, the electronic fiber was directly sewn in a piece of common compression bandage. Figure R10 (Supplementary Fig. 9a) shows photographs of the electronic fiber sewn into fabric and inside the collar of a shirt. One of the device units exposed in the spacing between threads of the fabric is measured. Therefore, the location of the device was marked using colored adhesive tape. However, for practical applications in a garment, an electrode connection at the end of the electronic fiber and between device units is required to access multiple devices independently, enabling the operation of multiple functions from a single fiber, hence overcoming the limitation of interconnection.

Figure R9. A photograph of the optoelectrical measurement when outside of the fiber device is irradiated by UV light.

Figure R10. Photographs of the electronic fiber sewn in a piece of a common compression bandage and then attached inside the collar of a shirt. (Supplementary Fig. 9)

In addition, to prevent unexpected breakdown and control the orientation of the fiber in the garment during sewing, we introduced a special sewing method (syringe needle assisted sewing method, See Video R1(Supplementary Video 1)). However, we have not achieved a proper measurement setup and unable to measure the optical signal through core. On the other hand, we successfully sewed individual electronic fiber in a garment using the syringe needle-assisted sewing method. Because the syringe needle guides the electronic fiber during sewing, we were able to control the orientation of the fiber in the garment, resulting in the UV sensor upward for both electrical contacts for UV sensing and uniform UV illumination.

According to the reviewer's comment, we carried out optoelectrical characteristics of the electronic fiber after integration to the collar of a shirt and after wearing for 10 days, as shown in Fig. R3 and Supplementary Fig. 9. The photocurrents of the electronic fibers were extracted from the transfer curves at $V_G = -5$ V and $V_D = 0$ V, under 470 nm LED light irradiation out of the fiber, which was the same measurement condition of Figure 3A in the revised manuscript. According to the reviewer's comments, we added the following paragraph on pages 14-15 and put Figure R3 as Fig. 5g in the revised manuscript.

- To prevent unexpected damage to the electronic fibre during the sewing process, we introduced a special sewing method by using a needle of a syringe that enables the fibres to place in the fabric safely (Supplementary Mov. 1). It helped us to sew the electronic fibre in the fabric without any serious damage to the surface. Because the syringe needle guides the electronic fibre during sewing, we were able to control the orientation of the fibre in the garment(bandage), resulting in the UV sensor or temperature sensor upward for both electrical contacts for UV or temperature sensing. **(on page 14)**
- We monitored the switching performance of the IGZO FET, optoelectrical characteristics of the UV sensor, and thermoelectrical characteristics of the temperature sensor on the fibre at pristine, sewed and after wearing for 10 days (Supplementary Fig. 9). The IGZO FET maintained its electrical performance for 1 day. After 10 days, it exhibited a slightly degraded field-effect mobility above $7.4 \text{ cm}^2\text{V}^{-1}\text{s}^{-1}$ (Fig. 5f). Additionally, we observed a slight decrease in drain current from 2.9 to 1.8 μA and a positive shift in threshold voltage from -2.4 to -1.7 V, respectively. From the additional measurements of the photocurrent and thermoelectric voltage of the electronic fibre before and after wearing for 10 days, as shown in Fig. 5g, we found that the photosensor was relatively stable, maintaining photocurrent values of 15.3 and 13.7 μA for the LED on and off states, respectively, at $V_G = 0$ V and $V_D = 5$ V. **(on page 14-15)**

5. Please comment more on the potential density of devices, limits to things such as track width, transistor feature sizes, even if they are just theory it would be good to know to see the potential for this methodology.

[Response] We thank the reviewer for this comment on the device density. We would like to show the feasibility of our electronic fiber platform as one of the potential emerging electronic fibers. As shown in Figure 1 of the original manuscript, devices can be spaced in fiber with a distance of only 25 μm . The length of the area of transistors (inverter) and ROs on fiber is 1 mm and 1.58 mm, respectively, achieving 30 device sets on a 10 cm-fiber, demonstrated as a proof of concept. To further increase device density, electrode contact pads in the device can be closely spaced with the active position of devices. Especially, device density is related to the dimension (size) of the device and electrode that is fabricated on the substrate. We have tried to scale down the devices as small as possible. Due to the

limitation of the laboratory-scale fabrication process, we were not able to do further scaling down of the dimension of the devices (e.g. transistor with $L = 10\mu\text{m}$ and $W = 25\mu\text{m}$ / Contact pads with $100\mu\text{m} \times 100\mu\text{m}$, $50\mu\text{m} \times 50\mu\text{m}$). If the semiconductor fabrication technology on a microfiber substrate is matured, it is possible to implement high-density electronic fibers similar to those of conventional semiconductor devices. Additionally, we did a theoretical calculation on how long fiber requires to integrate the first Pentium 200 MHz process that has released in 1996. It contained 3.3 million transistors, measured 90mm^2 , and was fabricated in a $0.35\mu\text{m}$ BiCMOS process with four levels of interconnect. If the same area of the chip is integrated on a circular microfiber with a diameter of $150\mu\text{m}$, it requires only 19.3 cm of microfiber to implement Pentium electronic fiber, as shown in Figure R11 (below).

Figure R11. Theoretical calculation on how long microfiber (diameter of $150\mu\text{m}$) requires to integrate the first Pentium 200 MHz process. (<https://www.intel.com/pressroom/kits/quickreffam.htm#pentium>, [https://en.wikipedia.org/wiki/Pentium_\(original\)#cite_note-3](https://en.wikipedia.org/wiki/Pentium_(original)#cite_note-3))

According to the reviewer's comment, we revised the following paragraph on page 5 in the revised manuscript.

- The assembly of multiple electronic systems on a microfibre, illustrated in Fig. 1a and 1b, consists of two different electronic parts: basic optoelectronic elements and a temperature sensor. The electronic components are integrated on the surface of a ten-centimetre-long cuboid shape monofilament with a diameter of $150\mu\text{m}$. As a proof of concept to implement direct assembly of electronic systems on a microfibre, our electronic fibre has relatively low integration density compared to conventional electronic systems on 2D wafers. However, by further scaling down each electronic part on a microfibre, it may be possible to implement high-density electronic fibres similar to those of conventional semiconductor devices. This implies that our electronic fibre platform can be considered one of the potential emerging electronic fibres. (on page5)

In addition, based on the reviewer's comment, it is worth mentioning the density of the electronic fiber in the revised manuscript. We added following sentences on page 5 in the revised manuscript.

- The microfibre substrate also features a 3D geometric shape including four planar faces throughout the length of the fibre that enables a higher integration density. The most efficient way to increase the density of electronic components is to use the entire circumference of fibre. In order to investigate the feasibility of this, we integrated electronic components on two different sides of the square sectional fibre. (**on page 5**)

Reviewer #3 (Remarks to the Author):

Comments to authors:

In the study, the capillary-assisted coating method and the maskless photolithography were implemented onto rectangular optical fiber (non-planar transparent substrate, a square-shaped microfiber made of fused silica) to fabricate quickly miniaturized functional electronic devices such as transistors, inverters, ring oscillators for data processing, as well as sensing or transducing units for detecting optical/thermal signals. Although the etching of optical fibers for sensing applications is not novel idea and studied a lot in the literature, the suggested assembly of multiple electronic system on the optical microfiber would enable new technological advances if the below most common major problems and challenges from wearability concept and textile manufacturing view point are solved and discussed for the emerging field of e-textiles in the paper:

-Optical fibers are not commonly preferred in textile and clothing manufacturing techniques due to the limitation of their mechanical properties. In the study, inorganic material-based microfiber is used (optical fiber is used), from the textile manufacturing view point and with regard to mass production, what if plastic microfiber is used instead of optical fiber? As it is known that polyamide, polyester, viscose etc. are well known common fibers used in textiles and clothing because of their mechanical properties such as easy to process in manufacturing and, comfortable and durable enough for wearer because of flexibility, tensile, extension properties etc. In case the plastic microfiber should be chosen as substrate instead of optical fiber as suggested in page 11, then the novelty and the contribution of the paper would be clear and easily explained. Can it be concluded that your method might be also valid for plastic microfibers or not? Any remark on this? Or how do you explain your contribution to literature then?

[Response] Thank you for the reviewer's kind comment. We also agree with the reviewer's comment. There are several technologies available for the fabrication of electronics on flexible substrates. M. Ahmed et al. demonstrated an ultrathin, flexible surface-mount sensor circuit [Ahmed, M. et al. *IEEE Sens. J.* 12, 864 (2012); Ahmed, M. and Butler, D. P. *J. Vac. Sci. Technol. B, Microelectron. Nanometer Struct.* 31, 050602 (2013)]. H. Lin et al. investigated the thermal and electrical conduction in the Au films deposited on the polyimide fiber, showing great potential in the flexible microelectronic device field as a flexible substrate [R4]. The glass transition temperature of the polyimide substrates employed, $\sim 400^\circ\text{C}$, does limit the processing temperature that can be used [Ahmed, M. and Butler, D. P. *J. Vac. Sci. Technol. B, Microelectron. Nanometer Struct.* 31, 050602 (2013)]. Polyimide, one of the flexible polymeric materials, has been used as flexible fiber substrates for the roll-type production of electronic fibers since it is an organic material with high performance, which possesses a series of great features, such as low thermal conductivity, high tensile strength, tensile modulus, thermal stability, chemical stability, radiation resistance, and insulativity [R4]. Therefore, the flexible polymeric filament can be considered as one of the substitutional fiber substrates for better flexibility and processability. As another example, we prepared a square-shaped filament of SU-8 that has a dimension of $100\ \mu\text{m} \times 100\ \mu\text{m}$. Due to the limited processing temperature of the SU-8 monofilament, we could not fabricate an IGZO-based field-effect transistor but prepared electrode patterns on the SU-8 filament. From the I - V and resistance characteristics of the metal electrode on the fiber substrate, we identify that there were negligible differences in electrical property between before (flat) and after bending (coiling it around a glass cylinder with a diameter of 1.3 mm), as shown in Figure R12. We conclude that the electrode on a plastic fiber well maintained its electrical property.

Figure R12. Photograph of the metal electrodes on the plastic fiber (a) flat and (b) coiled on a capillary tube with a diameter of 1.3 mm. (c) I-V characteristics and resistance of the metal electrodes on the plastic fiber on both flat and coiled.

We have revised the sentence on page 14 in the revised manuscript.

- Nevertheless, it is still required to develop better fibre substrate that enables both high performance and flexibility integrated electronic fibre system. For example, flexible polymeric materials, such as polyimide, include a combination of excellent properties, such as chemical stability, thermal stability, low thermal conductivity, radiation resistance, insulativity, high tensile strength and tensile modulus, which can be considered as substitutional fibre substrates for better flexibility and processability. [R4] (on page 14)

[R4] Lin, H. et al. Characterization of thermal and Electrical Transport in 6.4 nm Au films on polyimide film and fiber Substrates. *Sci. Rep.* **10**, 1–9 (2020)

Coating thick passivation (encapsulation) layer on the electronic fiber is a way to protect electronic fiber against various deformations. Because we have used a square silica core with a dimension of $150\ \mu\text{m} \times 150\ \mu\text{m}$, it requires hundreds of micrometers of the outer buffer layer to protect the fiber effectively by locating the fiber at the mechanical neutral plane [Kim, D. H. et al., *Adv. Mater.* **22**, 2108–2124 (2010)]. To explain the effectiveness of the outer protective layer, a knot was made using bare optical fiber to indirectly check how effectively the thick outer layer protects the fiber. As shown in Figures R13 and R14, the bare optical fiber has thick outer protective layers with thicker than $150\ \mu\text{m}$, exhibiting that the fiber was not broken under a bending radius of 5 mm (the calculated strain $\varepsilon = 5.0\text{--}5.5\%$). From the above results, we conclude that both plastic fiber and the introduction of a protective layer should be considered for practical use of electronic fiber with better reliability. Although we did not take into consideration a thick passivation or shielding layer on the outer of the electronic fiber at the present stage due to difficulty in the probe, we are fully aware of the necessity of encapsulation of the fiber for practical wearable application.

Figure R13. The structure of the optical fiber. The optical fiber features a $150\ \mu\text{m} \times 150\ \mu\text{m}$ of square silica core, $250\ \mu\text{m}$ of hard polymer cladding diameter, and $500\ \mu\text{m}$ of Tefzel buffer coating diameter. Note that this image is from <https://www.thorlabs.com/>

Figure R14. The optical fiber and its overhand knot.

-Moreover, the reliability and working range of miniaturized functional electronic devices onto optical fiber are not much discussed in the paper. The UV/thermal sensing behavior is not evaluated in detail. What are the reliability, sensitivity of the sensors, repeatability, hysteresis etc.? These issues should also be discussed.

[Response] Thanks for pointing this out. We have revised the following paragraphs in detail. (Page 9–10)

- Figure 3b presents the transfer characteristics of the IGZO-based FET on the fibre before and after UV exposure from out of fibre at $V_D = 5\ \text{V}$ with V_G sweeping from $-5\ \text{V}$ to $5\ \text{V}$. Upon illumination, there is a significant increase of off-current from $4.0 \times 10^{-8}\ \text{A}$ to $7.5 \times 10^{-7}\ \text{A}$ at $V_G = 0\ \text{V}$. This implies that exposed light contributes generation of photocarriers in the IGZO channel, inducing higher channel conductivity. Figure 3c displays time-dependent photoresponse at different gate voltages of $-1\ \text{V}$ and $0\ \text{V}$ with a drain voltage of $5\ \text{V}$ under pulsed illumination by UV light (power intensity: $1.0\ \text{mW cm}^{-2}$). (**on page 9**)
- The photocurrents of the electronic fibre were extracted from the transfer curves at $V_G = -5\ \text{V}$ and $V_D = 0\ \text{V}$, and photoresponsivity (R_λ) was calculated by

$$R_\lambda = \frac{I_{\text{light}} - I_{\text{dark}}}{P_{\text{opt}} \times A}$$

where I_{light} and I_{dark} are the drain current under light illumination and dark conditions, respectively; P_{opt} and A represent the incident illumination power (1.0 mW cm^{-2} for 470 nm LED light, $P_{opt} = 84.5 \text{ } \mu\text{W cm}^{-2}$ for 404 nm laser light, the illumination power was measured by a power metre) and effective area, respectively. A is the channel area (Width \times Length = $2 \times 10^{-6} \text{ cm}^2$) of the device. (on page 10)

The detailed discussion about thermal sensing behavior (Fig. R15) is described in Supplementary Fig. 6. in the revised supplementary information.

-
- **Figure R15. Thermoelectrical characteristics of temperature sensor on a microfibre.** Thermoelectric voltage (ΔV_{TE}) curves versus temperature gradient (ΔT) applied to (a) Chromium (Cr) and (b) Nickel (Ni) thin films. (c) Current (I) – voltage (V) curves and the calculated resistances of each thermocouple consisting of Cr and Ni thin film fabricated on the square-shaped glass fibre. (d) A plot of thermoelectric voltage versus time for the Cr–Al₂O₃–Ni thermocouples measured during a gradual increase in temperature. (e) Thermoelectric voltage as a function of temperature difference. Seebeck coefficient (S) of chromium (Cr) and nickel (Ni) thin films were measured with a homemade setup as described in our previous reports.² To determine S , it is necessary to apply a temperature

gradient (ΔT) across the sample and measure the thermoelectric voltage (ΔV_{TE}) and ΔT . Fine chromel and alumel wires (ϕ 50 μm , Nilaco) were used to resolve these problems. Chromel and alumel are alloys with known S and well-known as materials for forming K-type thermocouples. When ΔT is applied to the sample, the temperatures at the hot and cold sides can be measured at the same time in addition to ΔV_{TE} generated from the sample. A typical ΔV_{TE} versus ΔT curve when measuring Cr and Ni thin films (3 mm of width, 23 mm of length, 30 nm of thickness) are shown in **Fig. S6(a)** and **S6(b)**. From the above equation, S_{χ} can be evaluated using either the chromel-chromel or alumel-alumel wire pair or both. Here, $S_{\chi, \text{Chr-Chr}}$ and $S_{\chi, \text{Alu-Alu}}$ are the S_{χ} estimated through the inner two chromel-chromel and alumel-alumel wire pairs, respectively.

$$\begin{aligned} S_{\chi, \text{Chr-Chr}} &= \frac{\Delta V_{\text{Chr-Chr}}}{\Delta T} + S_{\text{Chr}}, \\ S_{\chi, \text{Alu-Alu}} &= -\frac{\Delta V_{\text{Alu-Alu}}}{\Delta T} + S_{\text{Alu}}, \\ S_{\chi} = S_{\chi, \text{Chr-Chr}} = S_{\chi, \text{Alu-Alu}} &= \frac{\Delta V_{\text{Chr-Chr}} - \Delta V_{\text{Alu-Alu}}}{2\Delta T} + \frac{S_{\text{Chr}} + S_{\text{Alu}}}{2} \end{aligned}$$

$15.4 \pm 0.1 \mu\text{V K}^{-1}$ and $-13.3 \pm 0.1 \mu\text{V K}^{-1}$ were obtained as S of Cr and Ni thin films ($S_{\text{Cr, film}}$ and $S_{\text{Ni, film}}$), respectively, from the linear slopes of the line fit to each ΔV_{TE} - ΔT curves and the equation, which matches well as S of thermocouple consisting of Ni and Cr ($S_{\text{Cr, film}} - S_{\text{Ni, film}}$) as discussed below. The S values of other reference samples were also measured to further confirm the accuracy of the measurement system. The previously reported S of chrome, gold, nickel, and platinum was $21.9 \pm 0.2 \mu\text{V K}^{-1}$, $2.0 \pm 0.4 \mu\text{V K}^{-1}$, $-19.3 \pm 0.4 \mu\text{V K}^{-1}$, and $-0.9 \pm 0.65 \mu\text{V K}^{-1}$, respectively. More detail about the schematic of the system, reference measurement, and theory can be found in previous reports.²

The quantity of heat flow (Q) across a temperature gradient can be expressed by *Fourier's* law as,

$$Q = -A\kappa \frac{\partial T}{\partial L} \quad (\text{Equation 3: Fourier's law}),$$

where A is the cross-section area of the material, κ is the material's thermal conductivity, T is the temperature gradient as a function of the distance from a thermal source (L), as shown in **Fig 4B**. As discussed in **Fig. S6(d)** and **S6(e)**, T at each TC junctions can be reversely calculated by using S of each TCs and ΔV_{TE} generated from each TCs, (See **Equation 4**).

$$T = \frac{\Delta V_{TE}}{S_{\text{Cr}} - S_{\text{Ni}}} + T_{\text{RT}} \quad (\text{Equation 4})$$

T at each TCs as plotted in **Fig. 4C** was converted from ΔV_{TE} and **Eq. 4** and showed a lower temperature in order away from the thermal source ($T_{\text{Source}} > T_{\text{TC-1}} > T_{\text{TC-2}} > T_{\text{TC-3}}$). The temperature distribution from the heat source at each sensor, as described in **Fig. 4D**, showed exponential decay due to heat loss from air convection instead of linear decay following *Fourier's* law. The temperature output of the thermoresistive TC column sensors fabricated on the 1D fibre indicates that it can be applied as a distributed temperature sensor.

-What about the abrasion during the production and usage? Fibers embedded to textile system will be definitely subjected to friction. Any results about this issue?

[Response] This is very important to commercialize our miniaturized functional electronic devices onto an optical fiber. Our electronic fiber is very vulnerable to scratches in their current state. To integrate various electronic components (transistor, inverter, ring-oscillator, and temperature sensor), it required a lot of repeated fabrication steps (CTAC process, maskless lithography, development,

etching, sputtering, thermal evaporation, etc.) We experienced many failures that are originated from unexpected abrasion of both semiconductor and metal layers during the production. The main reason of abrasion during device integration was scratches of the microfiber surface by unexpected physical contact with the inner wall of the capillary tube or positioning of the microfiber on the sample holder. Therefore, we had to do the fabrication process of the electronic fiber very carefully. The most important point on friction is expected to be happened during embedding of the fiber to the textile, as the reviewer's comment. We also recognized this issue and introduced a special sewing method by using a needle of a syringe which enables the fibers to embed safely. We took a short movie on how to embed the fiber to the textile in **Video R1**. It helps sewing of the electronic fiber in the fabric without any serious damage on the surface. As shown in Fig. 5E, we used this method and carried out electrical characteristics of the electronic fiber embedded in the fabric and sewed inside the collar of a shirt. Concerning the abrasion during usage, we performed our long-term monitoring of performances of the electronic fiber in the fabric for 10 days. We did not observe any serious damage which remains on the surface of the electronic fiber after use. However, if the fiber is exposed to physically harsh conditions under repeated bending with closed packing in the fabric, it is expected that the fiber will experience an abrasion of electrode or semiconductor layers that meet the fabric. Therefore, it is necessary to introduce surface passivation or a protective layer that can improve the durability of the electronic fibers in the fabric.

Concerning this, we added the following sentences on page 14 in the revised manuscript.

- To prevent unexpected damage to the electronic fibre during the sewing process, we introduced a special sewing method by using a needle of a syringe that enables the fibres to place in the fabric safely (Supplementary Mov. 1). It helped us to sew the electronic fibre in the fabric without any serious damage to the surface. Because the syringe needle guides the electronic fibre during sewing, we were able to control the orientation of the fibre in the garment(bandage), resulting in the UV sensor or temperature sensor upward for both electrical contacts for UV or temperature sensing. **(on page 14)**

Also, we put the following sentence on page 25 in the METHODS section in the revised manuscript.

- To sew electronic fibre in a fabric, a special sewing method was developed by using a needle of a syringe that enables the fibres to place in the fabric safely (Supplementary Video 1). **(on page 25)**

-Any extension and elongation results?

[Response] We appreciate the reviewer's comment. The glass fiber substrate is relatively flexible compared to glass-based planar substrates owing to its thinness and small width. However, the glass microfiber is prone to fracture under high mechanical stress. In fact, the glass has a tensile modulus of ~69 GPa, which is an order of magnitude larger than those of plastic materials [Antunes, P. et al., Mechanical Properties of Optical Fibers, *Selected Topics on Optical Fiber Technology*, Rijeka, Croatia:InTech, (2012)]. The inherent mechanical property of the glass fiber leads to limited flexibility with restricted motion. The maximum bendability of our electronic fiber was measured through addition bending tests to the limits, as shown in Figure R9. We found that the threshold bending radius (R) of our electronic fiber was measured to be 7.3 mm, implying the calculated bending strain of 1.03%. As the bending radius (R) reached right before 7.3 mm, it still kept its flexibility. It means that the electronic fiber return to its pristine state as shown in Figure R9A. As shown in original Figure 5a, b, and c, the electronic components on the glass fiber kept their electrical characteristics up to a strain of 0.68 % (Bending radius (R) of fiber: 11.0 mm). We carried out electronic characteristics of the IGZO TFT under different bending conditions ($R=\infty$, 13 mm and 7.3 mm) at the same time. The calculated strain (ϵ) values were 0.58 and 1.03 % at $R = 13$ and 7.3 mm, respectively. As shown in Figure R1E-G, there were negligible changes in the electrical performance including field-effect mobility, on/off ratio, and threshold voltage. However, the electronic fiber

(silica core) was broken when the bending radius (R) was less than 7.3 mm. Nevertheless, we would like to highlight that our 1D device has reasonable mechanical stability of the device performance under repeated bending cycles, even after 10,000 bending cycles at a fixed bending radius ($R = 11.7$ mm) (Fig. 5d). We think that the flexibility may still further improve in a future study by replacing both the substrate and the electrode with soft materials.

For better understanding of the mechanical properties of our electronic fiber, we added Figure R16 as Fig. S8. in the revised supplementary information and added following sentences on page 13 in the revised manuscript.

- From the additional mechanical tests and its electrical characteristics of the electronic fibre to the limit, we found that the threshold bending radius (R) and strain (ϵ) of the electronic fibre were 7.3 mm and 1.03%, respectively, without the mechanical failure of the silica fibre. (Supplementary Fig. 8) Because the electronic performance of the IGZO TFT on the fibre were maintained right before the breakdown of the fibre, the mechanical durability of our electronic fibre seems to depend on the mechanical properties of the fibre substrate. Due to the limitation of the mechanical properties of inorganic material-based microfibres, it was difficult to achieve better flexibility than that of conventional flexible electronic devices. However, it is essential to use high-performance inorganic semiconducting materials to implement high-performance and integrated electronic fibre system. Especially, silica-based microscale fibre can be considered as one of the promising fibre substrates due to its capability of achieving better electronic properties and scaling down by introducing both high-temperature and conventional semiconductor fabrication processes. Nevertheless, it is still required to develop better fibre substrate that enables both high performance and flexibility integrated electronic fibre system.

Figure R16. Maximum bending test of the electronic fiber and electrical characteristics of the IGZO

TFT on the electronic fiber. The bending radius (R) of (a) $R = \infty$, (b) $R = 13$ mm, (c) $R = 7.3$ mm, (d) $R < 7.3$ mm (Broken). The electronic fiber was broken at a bending radius of 7.3 mm. The calculated strain (ϵ) was 1.03 %. The electrical characteristics of the IGZO TFT on the electronic fiber at (e) $R = \infty$, (f) $R = 13$ mm, (g) $R = 7.3$ mm.

-The integration strategy of the fiber to PET substrate is not presented in detail. How did you do the sewing? Sewing is done manually or using a kind of sewing machine. I suppose the fiber can not be used in sewing machine because of low mechanical properties. Please give explanation.

[Response] We appreciate the reviewer’s kind comment. In order to carry out repeated bending tests as shown in Fig. 5D, we did not sew the fiber into the PET substrate and the electronic fiber was fixed on the PET substrate by using a polyimide tape, as shown in Figure R17. The polyimide tape enables solid attachment of electronic fiber on the PET for various bending conditions (e.g. tensile and compressive stress test, repeated bending test up to 10,000 cycles). (See Video R3)

Concerning the sewing of the electronic fiber, the electronic fiber was manually sewed by manually without any sewing machine. In order to prevent unexpected damage to the electronic fiber during the sewing process, we introduced a special sewing method by using a needle of a syringe which enables the fibers to place in the fabric safely (**Supplementary Video 1**). Although this is conceptual work, the weaving technologies for electronic fibers should be developed in a future study by practical wearable applications. Concerning this, we revised the following sentence on page 12 in the revised manuscript.

- The fibres were carefully placed and fixed on flexible polyethylene terephthalate (PET) substrates by using polyimide tape for both concave and convex bending conditions. (**on page 12**)

Figure R17. Schematics how to fix the electronic fiber on the PET substrate for repeated bending test.

-The working performance of the fiber in terms of all mentioned criteria (transistor, thermoelectrical characteristics of temperature sensors, optoelectrical characteristics of UV sensors etc.) should also be reported after integration to collar of a shirt and after wearing for some period.

[Response] Thank you for the reviewer’s kind comment. Wearable e-textiles should be breathable and washable with high flexibility and shape adaptability. To demonstrate feasibility for a potential e-textile application, the electronic fiber was directly sewed in a piece of common compression bandage. In order to prevent unexpected damage to the electronic fiber during the sewing process, we introduced a special sewing method by using a needle of a syringe which enables the fibers to place in the fabric safely (Supplementary Mov. 1). It helped sewing of the electronic fiber in the fabric without serious damage on the surface of the fiber. And then, the fabric was attached inside of the collar of a shirt again, without additional protective coating as shown in Fig. 5e and Supplementary Fig. 9.

According to the reviewer's comment, we carried out additional thermoelectrical and optoelectrical characteristics of the electric fiber after integration to the collar of a shirt and after wearing for 10 days, as shown in Fig. R18 and Supplementary Fig. 9. The photocurrents of the electronic fibers were extracted from the transfer curves at $V_G = -5$ V and $V_D = 0$ V, under 470 nm LED light irradiation out of the fiber, which was the same measurement condition of Figure 3a in the revised manuscript. Additionally, we performed thermoelectrical characteristics of each TC on the fiber in the fabric that was placed on a hot chuck of the probe station. As mentioned in the answer against reviewer #2, all TCs exhibited almost same value of around 0.52 μ V of thermoelectric voltage at room temperature (23 $^{\circ}$ C) before and after wearing for 10 days. Similarly, the thermoelectric voltage of each TC (TC1, TC2, TC3) showed a similar value of around 128.3 μ V at 40 $^{\circ}$ C before and after wearing for 10 days. We added the following sentences on page 14-15 and put Figure R18 as Fig. 5g in the revised manuscript.

- We monitored the switching performance of the IGZO FET, optoelectrical characteristics of the UV sensor, and thermoelectrical characteristics of the temperature sensor on the fibre at pristine, sewed and after wearing for 10 days (Supplementary Fig. 9). The IGZO FET maintained its electrical performance for 1 day. After 10 days, it exhibited a slightly degraded field-effect mobility above 7.4 $\text{cm}^2\text{V}^{-1}\text{s}^{-1}$ (Fig. 5f). Additionally, we observed a slight decrease in drain current from 2.9 to 1.8 μ A and a positive shift in threshold voltage from -2.4 to -1.7 V, respectively. From the additional measurements of the photocurrent and thermoelectric voltage of the electronic fibre before and after wearing for 10 days, as shown in Fig. 5g, we found that the photosensor was relatively stable, maintaining photocurrent values of 15.3 and 13.7 μ A for the LED on and off states, respectively, at $V_G = 0$ V and $V_D = 5$ V. Additionally, we evaluated the thermoelectrical characteristics of each TC on the fibre in the fabric placed on a hot chuck. All TCs exhibited almost the same value of approximately 0.52 μ V of thermoelectric voltage at room temperature (23 $^{\circ}$ C) before and after wearing for 10 days. Similarly, the average thermoelectric voltage of each TC (TC1, TC2, TC3) showed a similar value of approximately 128.3 μ V at 40 $^{\circ}$ C before and after wearing for 10 days. Although our electronic fibre worked well above stressful conditions, we should note here that an additional thick protecting or shielding layer is required to eliminate expected risks related to their contact with the human body while keeping the electrical function of the devices against various environmental conditions (mechanical stress, chemicals, sweat, etc.).
^{30,31} (on page 14-15)

Figure R18. Photocurrent of photosensor and thermoelectric voltage of TCs on the fiber at pristine, as sewed, and after wearing for 10 days.

Additionally, after the fiber is subjected to some environmental effects (e.g perspiration, abrasion) ? Please present results and give info. -Moreover, if it is washable or not, or should be detachable etc. please add info.

[Response] According to the reviewer's comment, we did additional experiments on the electrical characteristics of the transistors after washing to identify environmental effects.

At the first stage of the washing test, we prepared an electronic fiber without any passivation layer and fixed it on the side of the homemade holder, as shown in Figure R19. Then the electronic fiber was dipped in detergent solution (5 ml in 90 ml of tap water) for 30min and rinsed in pure tap water with stirring of 600rpm at room temperature. After washing, the electronic fiber was dried at 60 °C on a hotplate. As expected, the transistors on the glass fiber exhibited degradation of their performances before and after washing, as shown in Figure R20.

To further evaluate the washability of the electronic fiber, we carried out CTAC process (speed 1.0 mm min⁻¹) of SU-8 solution to form a passivation layer (thickness = 2 μm). The electronic fiber was completely covered by SU-8 passivation layer. Then, the encapsulated electronic fiber was dipped in detergent solution (5 ml in 90 ml of tap water) and NaCl solution (0.5 wt% for artificial human sweat) for 30 min and rinsed in pure tap water with stirring of 600 rpm at room temperature [Lim, T. et al., *Sci. Rep.* **9**, 17294 (2019)]. After washing, the electronic fiber was dried at 60 °C on a hotplate. Because SU-8 passivation completely covered the outer of the electronic fiber, the electronic performance of the IGZO FET showed a negligible difference in transfer characteristics at pristine and after washing with detergent and 0.5 wt% of NaCl solutions, as shown in Figure R21. It is well known that encapsulation (passivation) of the device is a general issue of the wearable electronic device that has to attach to the human body or garment. It is necessary to employ the passivation layer because it allows the protection of the device against external moisture, oxygen, and chemicals [S. J. Kim et al., *Advanced Materials*, **31**, 1900564 (2019)]. In addition, ambient species significantly degraded the performances of the metal oxide semiconductor-based thin-film transistors [D. Ho et al., *Journal of Material Chemistry C*, **8**, 14983-14995 (2020)]. Especially, the capability of encapsulation of the materials and their biocompatibility are essential for eliminating expected risks related to their contact with the human body and protecting the functionality of the electric devices [J. Kim et al., *Nature Biotechnology*, **37**, 389–406 (2019)]. For example, the encapsulated and protected devices could enhance safety by minimizing leakage currents and avoiding irritation/allergic reactions [X. Wang et al., *Small*, **13**, 1602790 (2017)]. To our best knowledge, the candidate materials for passivating or shielding 1D devices would be solution-processable organic insulating materials such as a cross-linkable elastomer, parylene-C, and hydrophobic polymers. Among various possible passivation layers, the solution-processable organic passivation layer is considered a good choice to employ the capillary tube assisted coating (CTAC) process for one-dimensional fiber substrate. Especially, SU-8 would be one of the promising passivation layers for IGZO-based thin-film transistors to minimize the impact of ambient species [J. Yoon et al., *Nature Communications*, **7**:11477 (2016); A. Olziersky et al., *Journal of Applied Physics*, **108**, 064505 (2010); D. Ho et al., *Journal of Material Chemistry C*, **8**, 14983-14995 (2020)].

According to the reviewer's comment, we added the following sentences on page 15-16 and added Figure R22 as Fig. 5h in the revised manuscript.

- To further evaluate the washability of the electronic fibre, we carried out the CTAC process (speed 1.0 mm min⁻¹) of SU-8 solution to form a passivation layer (thickness = 2 μm). The electronic fibre was completely covered by an SU-8 passivation layer. Then, the encapsulated electronic fibre was dipped in detergent solution (5 ml in 90 ml of tap water) and NaCl solution (0.5 wt% for artificial human sweat) for 30 min and rinsed in pure tap water with stirring at 600 rpm at room temperature. After washing, the electronic fibre was dried at 60 °C on a hotplate. Because the SU-8 layer completely covered the outer electronic fibre, the electronic performance of the IGZO FET showed a negligible difference in transfer characteristics (field-effect mobility of 3.74 cm²V⁻¹s⁻¹ in saturation regime and on/off current ratio of 4 orders of magnitude) before and after washing with detergent and 0.5 wt % NaCl solutions, as shown in Fig. 5h. This implies that the encapsulated electronic

fibre maintained a stable performance, regardless of the wet environment, such as washing and perspiration conditions. It may be possible to implement practical electronic fibre by introducing a reliable protecting or encapsulation layer that is durable under various mechanical or chemical conditions. (on page 15-16)

Also, we added the following sentences in the METHODS section on page 25-26 in the revised manuscript.

- To evaluate the washability of the electronic fibre, detergent (5 ml in 90 ml of tap water) and NaCl (0.5 wt% for artificial human sweat) were well dissolved in tap water. The washing test was performed for 30 min and rinsed in pure tap water with stirring at 600 rpm at room temperature. After washing, the electronic fibre was dried at 60 °C on a hotplate. (on page 25-26)

Figure R19. Photographs of washing test of the electronic fiber with and without SU-8 passivation layer in detergent solution and NaCl solution.

Figure R20. Electrical characteristics of the transistors on electronic fiber before and after washing (detergent solution) without passivation layer.

Figure R21. Electrical characteristics of the transistor on electronic fiber before and after washing (detergent and NaCl solution) with SU-8 passivation layer. SU-8 passivation layer was coated by CTAC process.

Figure R22. Electrical characteristics of the encapsulated IGZO FET on the electronic fiber before and after washing (detergent and NaCl solution).

-The square-shaped silica core microfiber (150 $\mu\text{m} \times 150 \mu\text{m} \times 7.5 \text{ cm}$), is it drawn in your lab or commercially available, please give info?

[Response] The silica microfiber is commercially available. The origin of the surface defects was strongly related to the preparation of the glass fiber which is the core of the optical fiber in this experiment. Specifically, to obtain the glass fiber (Pure silica core), we have to peel off all outer shells (e.g. Tefzel Buffer, Hard polymer cladding) as shown in Figure R23. The hard polymer cladding was

removed by chemical methods. However, the Tefzel Buffer was removed mechanically so that the fiber flaws seem to remain on the surface of the pure silica core during the preparation process.

According to the reviewer's comment, we already provided material information in METHODS section on page 24 in the revised manuscript

- Optical fibre with square-shaped silica core (FP150QMT, Thorlabs), ~ (on page 24)

Figure R23. The structure of the optical fiber. The optical fiber features a $150\ \mu\text{m} \times 150\ \mu\text{m}$ of square silica core, $250\ \mu\text{m}$ of hard polymer cladding diameter, and $500\ \mu\text{m}$ of Tefzel buffer coating diameter. Note that this image is from <https://www.thorlabs.com/>

-In the figures, generally schematic illustrations are given instead of real samples. It would be good to add real ones to convince the reader. Or any video demonstration would be also nice.

[Response] We thank the reviewer for this comment. We completely agree with the reviewer's comment. As the reviewer knows, we have added photographs and SEM images of real samples as shown in Figs. 1–5 and supplementary Figs. 1–3. Microscale devices on the fiber substrate are invisible to the naked eye, as shown in Fig. 1 and supplementary Fig. 8. We employ schematic illustrations to make them easier for readers to understand.

According to the reviewer's comment, we added photographs of measurement systems Figure R24 as Supplementary Fig. 5 in the revised supplementary information.

Figure R24. Photographs of the measurement setup. (a) Optoelectrical measurement when the UV laser is irradiated through the fiber core and is propagated within the single FET fabricated on the optical glass fiber. (b) Thermo-electrical measurement during detecting thermal information.

Minor Changes

1. The present address of Dr. Minji Kang have added on the first page of the revised manuscript as shown below.

“\$Present address: Chemical Materials Solutions Center, Korea Research Institute of Chemical Technology, 141 Gajeong-ro, Yuseong-gu, Daejeon 34114, Republic of Korea”

2. We added the copy of the Editing Certificate on our revised manuscript from Springer Nature Author Services (**The verification code 9DEA-083E-DA6F-8A79-6EDF**) below.

[REDACTED]

- End of response letter -

REVIEWERS' COMMENTS

Reviewer #1 (Remarks to the Author):

Review: Integration of multiple electronic components on a microfiber toward emerging electronic textile platform

The English language of the paper is not vetted.

Overall, this is a very interesting paper with large potential applications and benefit in e-textile development.

The comments of the three reviewers were generally understood, discussed, taken into account, and included in the paper.

Finally, the paper itself, the Supplementary Information document, and the other documents (notably videos) form a rich, complete, and comprehensible whole.

In view of the results presented and the corrections made, I recommend the publication as it stands.

Reviewer #2 (Remarks to the Author):

Many thanks for your revised manuscript. I believe you have answered all of my previous comments really well and, for me at least, made the paper significantly clearer and more relevant to the e-textile field. My only remaining comment would be to say that I really liked the Figure R11 you included in the rebuttal and I think it summarises the concept, achievement and future target really well in one simple figure so I would highly recommend you find a way to incorporate it in the main document.

REVIEWER COMMENTS

Reviewer #1 (Remarks to the Author):

The English language of the paper is not vetted.

Overall, this is a very interesting paper with large potential applications and benefit in e-textile development. The comments of the three reviewers were generally understood, discussed, taken into account, and included in the paper. Finally, the paper itself, the Supplementary Information document, and the other documents (notably videos) form a rich, complete, and comprehensible whole. In view of the results presented and the corrections made, I recommend the publication as it stands.

[Response] We appreciated the reviewer's kind comment.

Reviewer #2 (Remarks to the Author):

Many thanks for your revised manuscript. I believe you have answered all of my previous comments really well and, for me at least, made the paper significantly clearer and more relevant to the e-textile field. My only remaining comment would be to say that I really liked the Figure R11 you included in the rebuttal and I think it summarises the concept, achievement and future target really well in one simple figure so I would highly recommend you find a way to incorporate it in the main document.

[Response] Thank you for the reviewer's kind comment. We also agree with the reviewer's comment. To incorporate Figure R11, we carefully revised and put it as Figure 6 (see below) in the revised manuscript.

Again, we added the following sentences on pages 16-17 in the revised manuscript. "Due to the limitation of scaling down in the laboratory-scale fabrication process at the current status, we achieved 30 device sets (e.g. transistor with $L = 10\mu\text{m}$ and $W = 25\mu\text{m}$ / Contact pads with $100\mu\text{m} \times 100\mu\text{m}$, $50\mu\text{m} \times 50\mu\text{m}$) on a 10 cm long fibre, demonstrating as a proof of concept to implement direct assembly of electronic systems on a microfibre. If the semiconductor fabrication technology on a microfiber substrate is matured, we believe that it is possible to implement a higher density electronic fibre similar to those of conventional semiconductor devices on a silicon wafer. For more information on the feasibility of our electronic fibre platform, we calculated simply how long fibre requires to integrate a personal computer microprocessor (Intel Pentium processor) which has a die area of 91mm^2 and contains 3.3 million transistors with a process step of $0.35\mu\text{m}$ BiCMOS technology.³³ If the same fabrication technology is applied to integrate the above chip on the outer shell of the circular microfiber with a diameter of $150\mu\text{m}$, it requires only 19.3 cm long microfiber to implement microprocessor fibre, as shown in Figure 6."

And we put following figure caption of Fig. 6. on page 27 in the revised manuscript. Also, we put following link "<https://www.intel.com/pressroom/kits/quickreffam.htm#pentium>" as ref 33 in the revised manuscript.

"Fig. 6. Schematics of the implementation of the microprocessor on a microfibre. Theoretical calculation on how long microfiber (diameter of $150\mu\text{m}$) requires to integrate the Pentium microprocessor.³³"